# xCodeEval: An Execution-based Large Scale Multilingual Multitask Benchmark for Code Understanding, Generation, Translation and Retrieval

## Abstract

Recently, pre-trained large language models (LLMs) have shown impressive abilities in generating codes from natural language descriptions, repairing buggy codes, translating codes between languages, and retrieving relevant code segments. However, the evaluation of these models has often been performed in a scattered way on only one or two specific tasks, in a few languages, at a partial granularity (e.g., function) level, and in many cases without proper training data. Even more concerning is that in most cases the evaluation of generated codes has been done in terms of mere lexical overlap with a reference code rather than actual execution. We introduce xCodeEval, the largest executable multilingual multitask benchmark to date consisting of 25M document-level coding examples (16.5B tokens) from about 7.5K unique problems covering up to 11 programming languages with execution-level parallelism. It features a total of 7 tasks involving code understanding, generation, translation and retrieval. xCodeEval adopts an execution-based evaluation and offers a multilingual code execution engine, `ExecEval` that supports unit test based execution in all the 11 languages. To address the challenge of balancing the distributions of text-code samples over multiple attributes in validation/test sets, we propose a novel data splitting and a data selection schema based on the geometric mean and graph-theoretic principle. Our experiments with OpenAI's LLMs (zero-shot) and open-LLMs (zero-shot and fine-tuned) on the tasks and languages demonstrate xCodeEval to be quite challenging as per the current advancements in language models. Both xCodeEval [1] and `ExecEval` are freely available at *Hugging Face* [2] and Github [3].

## 1 Introduction

Automatically generating computer programs to solve complex problems has been a long-standing goal in AI (Manna and Waldinger, 1971). In recent years, specifically with the growth of large language models (LLMs), we have witnessed tremendous progress in synthesizing code that is not just relevant but also fully functional with no further human modification needed (Chen et al., 2021). The progress made in related tasks such as program synthesis (Chowdhery et al., 2022; Li et al., 2022), program repair (Berabi et al., 2021), code translation (Roziere et al., 2020; 2021), and code retrieval (Wan et al., 2019; Parvez et al., 2021) are having a profound impact on increasing developer productivity (Ziegler et al., 2022) and aiding educators (Finnie-Ansley et al., 2022).

Despite the fact that such advancements are expected to be general with proper benchmarks, their evaluation has often been performed in a scattered way on a limited number of languages such as *Python* and *Java*, on a partial granularity level such as at the level of a statement (Huang et al., 2022) or function (Husain et al., 2019), focusing on only one or two specific tasks such as program synthesis and translation, and in many cases without proper fine-tuning data (Austin et al., 2021) or in terms of

---

1. `https://github.com/ntunlp/xCodeEval`
2. `https://huggingface.co/datasets/NTU-NLP-sg/xCodeEval`
3. `https://github.com/ntunlp/ExecEval`

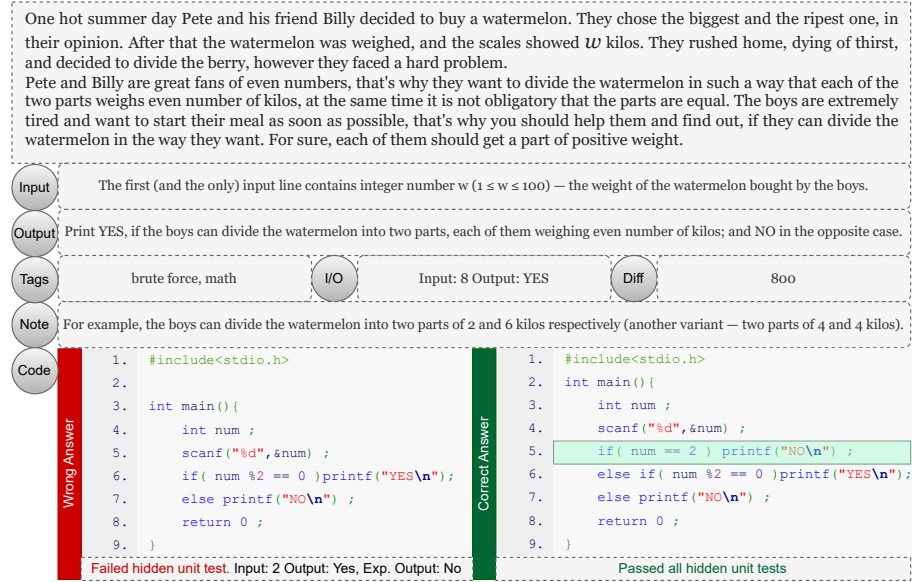

FIGURE 1 – A sample from XCODEEVAL. It includes a natural language description of the problem, input/output (i/o) description, and a few i/o examples. It also includes relevant meta-information such as problem tags (e.g., brute force, math), language, difficulty level (800 in the figure), and a note (explanation of i/o). Each sample contains a number of hidden unit tests (not shown in the figure) against which we evaluate the code. Although the code at the left gives the correct answer to the given input, it is incorrect as it fails in other test cases.

mere lexical $n$-gram based relevance (Iyer et al., 2018) rather than actual execution. We present a summary of the characteristics of existing program evaluation test-beds in Tables 1 and 6.

To address these limitations, and drive further advancements in the creation of more general-purpose LLMs for problem solving, we introduce XCODEEVAL, the largest executable multilingual multitask benchmark to date consisting of 20M coding examples from about 7.5K unique algorithmic problems. It covers up to 17 programming languages with the parallelism of multilingual data which can benefit both mono- and multi-lingual code intelligence applications. It features a total of 7 tasks involving code understanding, generation, translation and retrieval, and wherever appropriate it employs an execution-based evaluation protocol. A detailed documentation of the dataset can be found in Appendix (Section 7). Figure 1 shows an example from XCODEEVAL; it includes a problem description in natural language, a buggy and bug-free solution to the problem, and relevant metadata such as difficulty level, language, problem tags (e.g., brute force).

TABLE 1 – A comparison of the total number of unit test cases provided with the benchmark. Here ∞ means automated unit test generation by EvoSuite. N/A refers to unit tests not openly available. For our retrieval tasks, each candidate is pre-evaluated against the test cases.

| Benchmark | |La| | |Unit Test| |
|---|---|---|
| TransCoder (Roziere et al., 2020) | 3 | 14,100 |
| HumanEval (Chen et al., 2021) | 1 | 1,325 |
| HumanEval-x (THUDM, 2022) | 9 | 840 |
| MBPP (Austin et al., 2021) | 1 | 1,500 |
| TransCoder-ST (Roziere et al., 2021) | 3 | ∞ |
| APPS (Hendrycks et al., 2021) | 1 | 22,711 |
| MBXP (Athiwaratkun et al., 2022) | 10 | 1,500 |
| CodeContests (Li et al., 2022) | 3 | 27,220* |
| **XCODEEVAL** (ours) | | |
| – Classification tasks | 11 | - |
| – Generation tasks | 11 | 62,798 |
| – Retrieval tasks | 17 | 62,798 |

XCODEEVAL is a result of a number of crucial design principles and challenges as highlighted below.

**Reasoning** In terms of genre, *problem solving* posits a unique set of challenges that require (a) understanding a complex natural language problem description, (b) expertise in data structures and algorithms, (c) complex reasoning that goes beyond memorization, and (d) generating programs of potentially hundreds of lines so that they can pass a comprehensive list of especially designed hidden tests. Given the current progress in LLMs and their instruction following capability (Ouyang et al., 2022), competition-level problems that humans find challenging, provide an interesting benchmark to test many aspects of intelligence (Li et al., 2022; OpenAI, 2023).

**Multilinguality** We aim to cover as *many programming languages* as possible regardless of the resource discrepancies. One of the main objectives of this benchmark is to assess the degree to which codes in different languages are parallel to one another. In addition to that, we also intend to evaluate the zero-shot cross-lingual capability of the LLMs.

**Evaluation and its granularity** We believe the current evaluation standards do not fully consider the idea of the *global* meaning representation of a program, which requires models to comprehend different interpretable code segments and connect both local and modular knowledge into a global representation. We propose *execution-based* evaluation with unit tests at the global level. While there are many benchmarks covering the local understanding of a code segment, there are only a few that work at a global level as shown in Table 6. We consider a pair of codes to be equivalent, if they generate the same output for a given input regardless of syntax/languages (Sajnani, 2016). To support this, we have developed ExecEval, a new standardized and distributed execution environment that supports 44 compilers/interpreters in all the languages in XCODEEVAL . We also provide a large number of necessary unit tests (average of 50 per problem) for the relevant tasks (Table 1). In this context, it is noteworthy that 44 out of 165 problems in the CodeContest's test split have no private unit tests. Additionally, it contains 104 problems without complete collection of unit tests (as available in the source), thus are inadequate in assessing a solution's correctness. We have identified this issue and excluded such problems from our evaluation sets (development and test splits).

**Task difficulty and trainability** We wish to focus on problems of different difficulty levels (from 800 to 3500 rating points, following codeforces.com) such that models with different capabilities can be benchmarked against difficulty levels. We also aim to provide sufficient training data for each task so that pre-trained LMs can be fine-tuned or small-scale models can be trained from scratch.

**Data split** Finally, balancing the validation and test distributions of text-code instances over multiple attributes such as problems, tags, and execution outcome (e.g., correct vs. wrong) is challenging. We propose a novel data split schema based on a geometric mean and a data selection schema adapting a graph-theoretic solution to the circulation problem with lower and upper bounds (Mount, 2017) that can be applied for other benchmarks as well (Section 2.1).

We evaluate ChatGPT on our classification and generative tasks, and StarEncoder (Li et al., 2023) on the retrieval tasks. In addition to that we also trained *Program Synthesis* task on Starcoderbase-3B and compare it's result with CodeLlama-7b and CodeLlama-13b instruct model. Our results indicate that XCODEEVAL remains difficult to solve for the advanced LLMs, even on a simple binary classification task like Code Compilation (Table 3). With XCODEEVAL, we can identify and compare multilingual executablity across languages as well as perform open-ended evaluation on any programming language for the Code Translation and Program Synthesis tasks. Moreover, the unique parallelism of unit-tests allows for different interpretable evaluation and analysis on diverse code related tasks (Section 3). Finally, our experimental results with program synthesis tasks demonstrates that our training data can facilitate a reduction in the size of the language model while maintaining its executable capabilities.

## 2 XCODEEVAL: DATA, EXECUTION ENGINE & TASKS

### 2.1 DATA CREATION

We construct our dataset from 25M openly available samples from codeforces.com for a total of 7514 distinct problems. Each sample $S_k \in \mathcal{S}$ represents a potential solution to a problem $P_i \in \mathcal{P}$, and a problem $P_i$ can be solved by employing a set of algorithmic techniques $\mathbb{T}_i \subset \mathcal{T}$, which we refer to as problem tags (e.g., 2-sat, binary search); see Figure 8 for a complete list of tags.

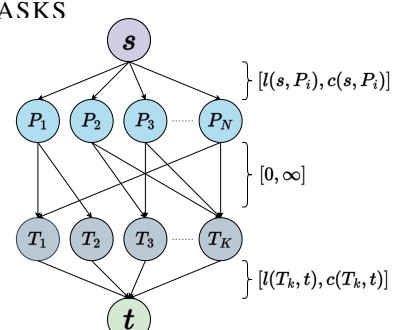

**Validation and test sets** To prevent data leakage, we first put aside $N_h (= 1354)$ problems as held-out set $\mathcal{D}_{\text{ho}}$ for validation and test. It ensures that the problems in the validation and test sets are not seen in training and models need to generalize to unseen problems. We then create $\mathcal{D}_{\text{valid}}$ and $\mathcal{D}_{\text{test}}$ splits from $\mathcal{D}_{\text{ho}}$, while maintaining a balanced tag distribution and ensuring that all the tags in these two sets also exist in the training

FIGURE 2 – Flow network of for validation-test dataset creation. Here $s$ and $t$ represent the source and sink of the flow network. Also, $l(u,v)$, $c(u,v)$ represents the lower and upper capacity of the edge connected from $u$ to $v$.

data, which could be a requirement for certain tasks (e.g., *Tag Classification*). For this, we iterate over a number of seeds and create random splits. Let $\gamma$ be the expected ratio of the number of samples in $\mathcal{D}_{\text{valid}}$ and $\mathcal{D}_{\text{test}}$, i.e., $\gamma = |\mathcal{D}_{\text{valid}}|/|\mathcal{D}_{\text{test}}|$. For each random split, we calculate a tag-wise ratio $\gamma_T$, the ratio of the number of samples in $\mathcal{D}_{\text{valid}}$ and $\mathcal{D}_{\text{test}}$ for each tag $T \in \mathcal{T}$. The geometric mean of $\{\gamma_T\}_{T \in \mathcal{T}}$ defines the 'tag distribution' score of a split. We select the split whose score is closest to $\gamma$. Appendix C-Algorithm 1 describes our method, which also ensures that the validation and test sets contain the same tag sets as the training set.

Next, to make the testing/validation computationally feasible, we aim to control the sample size while maintaining a balanced distribution across problems and tags; e.g., only C++ initially had about 647K test samples for tag classification (Appendix E.1). However, finding an optimal solution to this selection problem (i.e., how many samples to select per problem and per tag) is nontrivial. We formulate this as a *circulation problem with lower and upper bounds* (Mount, 2017) within a flow network. Let $p_i$ and $t_k$ be the number of solutions for a problem $P_i$ and a tag $T_k$, respectively. Let $G = (V, E)$ be a flow network (a directed graph) with the set of vertices $V = \{s, P_1, ..., P_N, T_1, ..., T_K, t\}$, where $s$ and $t$ respectively denote the source and sink nodes of the network (Figure 2). For each edge $e \in E$, the lower capacity $l(e)$ and upper capacity $c(e)$ are defined as follows.

1. Initialize $E = \emptyset$.

2. For each problem $P_i$, add edge $(s, P_i)$ to $E$ and assign $l(s, P_i) = \min(m_p, p_i)$ and $c(s, P_i) = \min(x_p, p_i)$, where $m_p$ and $x_p$ respectively refer to the minimum and maximum samples to choose per problem if available with $m_p \leq x_p$, thus $0 \leq l(s, P_i) \leq c(s, P_i)$.

3. For each tag $T_k$, add edge $(T_k, t)$ to $E$ and assign $l(T_k, t) = \min(m_t, t_k)$ and $c(T_k, t) = \min(x_t, t_k)$, where $m_t$ and $x_t$ respectively refer to minimum and maximum samples to choose per tag if available with $m_t \leq x_t$, thus $0 \leq l(T_k, t) \leq c(T_k, t)$.

4. For each $P_i$ and $T_k$, add $(P_i, T_k)$ to $E$ if $P_i$ has a tag $T_k$, and assign $l(P_i, T_k) = 0$, $c(P_i, T_k) = \infty$.

We then directly adopt the circulation problem solution to find a flow $f : E \longrightarrow \mathbb{Z}_+$ [4] that satisfies: $\forall e \in E, \; l(e) \leq f(e) \leq c(e)$ and $\forall u \in V, \sum_{v \in V} f(u, v) = 0$. In our case, $f$ denotes a feasible flow when the above constraints are satisfied for some $G$. For each $e \in E$, $f(e)$ represents the following:

1. $f(s, P_i)$ denotes the number of samples to be picked from problem $P_i$.

2. $f(T_k, t)$ denotes the number of samples to be picked from tag $T_k$.

3. $f(P_i, T_k)$ denotes the number of samples to be picked from $P_i$ that has a tag $T_k$.

Here, $\sum_{k=1}^{K} f(T_k, t) = \sum_{i=1}^{N} f(s, P_i)$ is the total number of samples selected, which can be controlled in a balanced way by setting the control variables $m_p$, $m_t$, $x_p$, and $x_t$. Appendix D gives further details about the method and hyperparameters for different tasks, along with a comparison to a random data selection strategy.

## 2.2 EXECEVAL: A MULTILINGUAL, DISTRIBUTED AND SECURED EVALUATION ENGINE

An essential requirement for execution-based evaluation is the availability of a secure and scalable framework (Chen et al., 2021; Cassano et al., 2022). With its capacity to support 44 compiler/interpreter versions in 11 different languages, ExecEval offers a versatile and comprehensive approach to program evaluation. The engine is distributed as a secure Docker image, ensuring safe and efficient executions. It also supports easy integration of new compilers/interpreters with custom execution flags (which can also be changed at run-time). While running on unit tests, ExecEval produces one of the six outcomes: (i) COMPILATION ERROR: fails to compile or run due to a syntax error; (ii) RUNTIME ERROR: successfully compiles but fails during runtime due to native environment issues (e.g., asserts, division-by-zero); (iii) MEMORY LIMIT EXCEEDED: occupies more memory than the limit; (iv) TIME LIMIT EXCEEDED: requires more time than the limit; (v) WRONG ANSWER: successfully compiles/interprets, generates an output but fails to produce a correct answer; (vi) PASSED: successfully passes all the unit tests. The program will be flagged as *buggy* (i-v) even when it fails on a single unit test. Appendix H of supp. material gives further details about ExecEval.

---

4. $\mathbb{Z}_+$ denotes the set of non-negative integers.

TABLE 2 – Dataset statistics per language and task (except retrieval). The validation and test splits for *Program Synthesis* are same across all the languages as they solve the same problems. *Code Translation* data refers to the source language. Since our setup supports execution-based evaluation, both *Program Synthesis* and *Code Translation* support any number of languages that are supported by the execution framework `ExecEval`.

| Split | C | C# | C++ | Go | Java | Javascript | Kotlin | PHP | Python | Ruby | Rust | Total |
|---|---|---|---|---|---|---|---|---|---|---|---|---|
| **Tag Classification** | | | | | | | | | | | | |
| Train | 178,324 | 79,128 | 3,711,550 | 25,608 | 703,625 | 15,716 | 49,340 | 6,234 | 678,576 | 15,226 | 30,681 | 5,494,008 |
| Validation | 1,694 | 2,234 | 1,983 | 1,626 | 1,908 | 1,610 | 1,712 | 891 | 1,969 | 2,149 | 920 | 18,696 |
| Test | 6,193 | 6,020 | 9,720 | 6,504 | 8,881 | 6,431 | 6,841 | 3,598 | 8,195 | 8,671 | 3,679 | 74,733 |
| **Code Compilation** | | | | | | | | | | | | |
| Train | 503,458 | 170,407 | 15,147,814 | 53,561 | 2,007,940 | 36,949 | 104,970 | 18,099 | 1,793,141 | 26,362 | 52,449 | 19,915,150 |
| Validation | 1,000 | 1,000 | 1,000 | 212 | 1,000 | 454 | 482 | 102 | 1,000 | 50 | 94 | 6,394 |
| Test | 5,000 | 5,000 | 5,000 | 814 | 5,000 | 1,676 | 1,940 | 392 | 5,000 | 242 | 324 | 30,388 |
| **Program Synthesis** | | | | | | | | | | | | |
| Train | 179,508 | 79,681 | 3,744,367 | 25,753 | 707,603 | 15,916 | 51,831 | 6,334 | 681,780 | 15,336 | 30,732 | 5,538,841 |
| Validation | 106 | 106 | 106 | 106 | 106 | 106 | 106 | 106 | 106 | 106 | 106 | 106 |
| Test | 952 | 952 | 952 | 952 | 952 | 952 | 952 | 952 | 952 | 952 | 952 | 952 |
| **Code Translation (Source Language)** | | | | | | | | | | | | |
| Train | 179,508 | 79,681 | 3,744,367 | 25,753 | 707,603 | 15,916 | 51,831 | 6334 | 681,780 | 15,336 | 30,732 | 5,538,841 |
| Validation | 768 | 746 | 1,054 | 470 | 960 | 412 | 421 | 374 | 868 | 494 | 467 | 7,034 |
| Validation small | 40 | 40 | 40 | 40 | 40 | 40 | 40 | 40 | 40 | 40 | 40 | 440 |
| Test | 1,725 | 1,760 | 1,981 | 1,811 | 1,849 | 1,651 | 1,949 | 1,734 | 1,942 | 1,928 | 2,026 | 20,356 |
| **Automatic Program Repair or APR** | | | | | | | | | | | | |
| Train | 135,307 | 37,039 | 3,409,220 | 13,085 | 574,448 | 8,861 | 16,338 | 3,595 | 461,356 | 5,153 | 7,668 | 4,672,070 |
| Validation | 733 | 739 | 641 | 294 | 716 | 183 | 313 | 191 | 710 | 343 | 205 | 5,068 |
| Test | 1,957 | 2,002 | 2,026 | 1,427 | 2,032 | 643 | 1,978 | 1,156 | 2,012 | 1,561 | 905 | 17,699 |

## 2.3 TASKS IN XCODEEVAL

XCODEEVAL features two classification, three generative, and two retrieval tasks. Table 2 gives a breakdown of the classification and generative tasks per language. Below we briefly describe the tasks; detailed **descriptions**, **motivation**, **maintainance, support**, and process of **task formulation** along with **visualizations** of task distributions and **task creation rationale** can be found in Appendix E.

**Classification tasks – Tag Classification and Code Compilation**    The goal of *Tag Classification* is to assign relevant tags to a code and/or natural descriptions of the corresponding problem. This task focuses on measuring the impact of code understanding by incorporating a natural language description alongside the code. It is the only task in our benchmark that does not factor in the code's executability. On the contrary, the objective of the *Code Compilation* task is to determine whether the given code is compilable or not, thereby constituting a binary classification problem. All the labels in both tasks are human annotated found as meta data. By addressing these classification tasks, we aim to explore and evaluate the effectiveness of program comprehension techniques.

**Generative tasks – Program Synthesis, Automatic Program Repair (APR) and Code Translation** All of our proposed generative tasks are evaluated with execution-based unit tests by `ExecEval`. The *Program Synthesis* task aims to generate executable programming language code that solves a given problem. The problem is defined with a natural language description along with some sample input-output descriptions (see Figure 1). In the *APR* task, along with the problem, a buggy code is also given. The objective is to *correct* or *refine* the buggy code. Moreover, in the *Code Translation* task, a code is provided in a source language and the goal is to translate it to a target language. Note that for *Code Translation* our benchmark provides the inputs for the source programming language and for *Program Synthesis* we only provide problem description in natural text. For both *Program Synthesis* and *Code-Translation*, the underlying execution-based unit test enables evaluation on any target programming language, as long as it is supported by `ExecEval`.

**Retrieval Tasks – Code-Code and NL-Code Retrieval**    The objective of the *Code-Code* retrieval is to retrieve relevant *executable* code when provided with a programming language code as input. On the contrary, the *NL-Code* retrieval task aims to retrieve relevant executable code based on a problem description. These retrieval tasks are novel in the sense that they consider both the *relevance* and *executability* of the retrieved codes for evaluation. To the best of our knowledge, these are the first retrieval-based tasks that incorporates *executability* as a crucial factor when performing code retrieval. We have also included a retrieval corpus specific to each of the languages for evaluation purposes.

## 3 EVALUATION AND ANALYSIS

For all tasks except *Code Translation*, we evaluate on the validation split. For *Code Translation* from source languages, we used the small validation split (follow Appendix E.4 in supp. material).

TABLE 3 – Performance of **gpt-3.5-turbo** on XCODEEVAL : For *Code Translation*, *X*- denotes the case where the source language is *X*, and the target languages are represented by the respective columns. For program synthesis, (T) denotes sampling done at 20 different temperatures ranging from 0.0 to 2.0, while (N) denotes sampling done at a fixed temperature 0.32 (see Section 3.2).

| Tasks | metric | C | C# | C++ | Go | Java | Javascript | Kotlin | PHP | Python | Ruby | Rust | AVG |
|---|---|---|---|---|---|---|---|---|---|---|---|---|---|
| TC-*Code2Tag* | macro-F1 | 32.37 | 26.91 | 40.58 | 23.06 | 31.58 | 19.35 | 33.95 | 15.25 | 29.45 | 23.64 | 24.04 | 27.29 |
| TC-*DesCode2Tag* | macro-F1 | 36.05 | 33.18 | 47.1 | 31.5 | 38.26 | 27.81 | 39.61 | 19.36 | 33.73 | 30.61 | 32.35 | 33.6 |
| Code Compilation | accuracy | 65.9 | 54.9 | 58.0 | 70.28 | 53.0 | 65.64 | 56.64 | 76.47 | 70.9 | 70.0 | 54.26 | 63.27 |
| Program Synthesis (T) | pass@5 | 25.37 | 30.59 | 31.36 | 31.03 | 29.74 | 22.74 | 26.87 | 30.17 | 33.98 | 33.72 | 10.28 | 27.8 |
| Program Synthesis (N) | pass@5 | 31.23 | 30.78 | 35.44 | 30.58 | 31.52 | 28.63 | 27.38 | 32.13 | 29.77 | 29.66 | 28.2 | 30.48 |
| Automatic Program Repair | pass@5 | 44.32 | 53.38 | 28.88 | 65.95 | 33.21 | 86.05 | 62.49 | 64.22 | 37.94 | 60.38 | 68.96 | 55.07 |
| Translation C-{} | pass@5 | - | 41.74 | 89.44 | 49.73 | 57.81 | 30.94 | 37.49 | 44.43 | 45.67 | 35.14 | 51.92 | 44.03 |
| Translation C#-{} | pass@5 | 62.27 | - | 72.14 | 49.27 | 63.94 | 25.49 | 44.39 | 60.22 | 62.62 | 68.84 | 62.16 | 51.94 |
| Translation C++-{} | pass@5 | 49.78 | 48.47 | - | 49.43 | 48.98 | 22.65 | 33.91 | 35.41 | 31.88 | 39.46 | 39.1 | 36.28 |
| Translation Go-{} | pass@5 | 59.72 | 63.75 | 79.18 | - | 69.92 | 51.46 | 25.2 | 36.05 | 71.05 | 42.81 | 51.21 | 50.03 |
| Translation Java-{} | pass@5 | 46.03 | 28.13 | 52.64 | 46.82 | - | 32.21 | 28.5 | 11.53 | 44.38 | 27.07 | 42.16 | 32.68 |
| Translation Javascript-{} | pass@5 | 57.64 | 49.16 | 68.04 | 64.49 | 60.24 | - | 16.1 | 31.52 | 64.12 | 14.93 | 52.27 | 43.5 |
| Translation Kotlin-{} | pass@5 | 74.34 | 59.39 | 85.67 | 51.52 | 39.2 | 29.01 | - | 39.43 | 64.58 | 53.33 | 53.97 | 50.04 |
| Translation PHP-{} | pass@5 | 64.38 | 17.5 | 55.92 | 62.19 | 52.11 | 26.19 | 2.5 | - | 59.79 | 64.33 | 36.87 | 40.16 |
| Translation Python-{} | pass@5 | 41.18 | 19.38 | 42.58 | 50.82 | 40.65 | 19.93 | 6.04 | 48.69 | - | 68.12 | 22.23 | 32.69 |
| Translation Ruby-{} | pass@5 | 30.47 | 5.63 | 35.69 | 67.01 | 40.07 | 5.69 | 3.75 | 58.87 | 67.28 | - | 12.23 | 29.7 |
| Translation Rust-{} | pass@5 | 39.49 | 44.72 | 54.29 | 44.6 | 57.5 | 36.24 | 20.43 | 37.91 | 51.32 | 51.17 | - | 39.79 |
| Target lang. Avg | pass@5 | 52.53 | 37.79 | 63.56 | 53.59 | 53.04 | 27.98 | 21.83 | 40.41 | 56.27 | 46.52 | 42.41 | 45.08 |

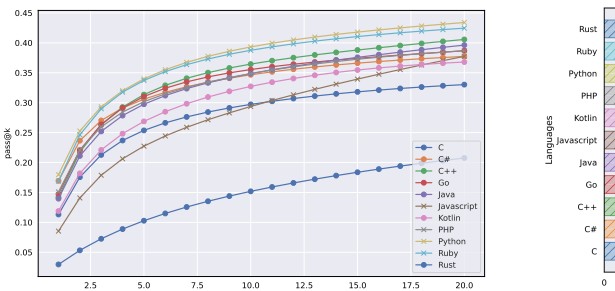
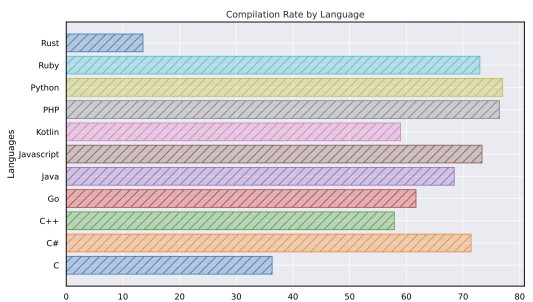

FIGURE 3 – Left: pass@$k$ for different languages at different $k$. Right: ratio of the number of generated codes that compiles for different languages. Both evaluations are done at 20 different temperature values.

## 3.1 BENCHMARK RESULTS

**Baselines** We benchmark XCODEEVAL using ChatGPT (*gpt-3.5-turbo-0301*). To construct a query prompt, we adopt the direct zero-shot prompting method (i.e., no chain-of-thought) that facilitates easier automated evaluation (no overlapping of code and explanations). For the retrieval task, following (Karpukhin et al., 2020), we build a bi-encoder based dense multilingual code retriever by finetuning the *StarEncoder* (Li et al., 2023) model. Our implementation details are provided in Appendix G.

**Results** We present the results on the classification and generative tasks in Table 3. Overall, the model achieves inspiring yet inadequate results – marking XCODEEVAL a promising yet challenging benchmark as per the current progress in LLMs. Particularly for *Tag Classification*, we observe decent performance in general and incorporating a problem description enhances the model's overall predictive capability. However, in web languages (e.g., *PHP*, *JavaScript*), it exhibits the poorest performance. In *Code Compilation*, we observe encouraging performance for *Go*, PHP, Python, *Ruby*, and *C#*. However, the performance is close to a random baseline for Java, *Kotlin*, and *Rust*.

For *Program Synthesis*, we find that in popular languages, such as *C++*, C#, Go, Java, and Python the model performs well, while in rare languages like Rust it fares poorly. Notably while on other datasets such as HumanEval (Chen et al., 2021), ChatGPT achieves much higher scores, 65.8 in pass@1 (OpenAI, 2023; Chen et al., 2022), it significantly falls behind even in pass@5 ($\sim 30$) in XCODEEVAL – imposing challenges even for such a powerful LLM. In Figure 3 (left), we show the performance for different $k$. As expected, results increase with $k$ and no noticeable differences are observed between the compiler (e.g., C++, Java) and interpreter (e.g., Python, Ruby) languages.

In *APR*, we observe a higher performance scale than in *Program Synthesis* indicating that the model finds the task relatively easier since it does not necessitate generating a complete code from scratch. Interestingly, in languages where the model underperformed in program synthesis, it exhibits good

TABLE 4 − Summary of the performance of *StarEncoder* Li et al. (2023) finetuned on our retrieval tasks for $k = 100$. For *Code-Code*, ($\alpha$) denotes the average score for codes of any given language as the corpus, similarly ($\gamma$) denotes average score for codes of any fixed language as query. For *NL-Code*, the scores are reported for corpus of different languages.

| Tasks | metric | C | C# | C++ | D | Go | Haskell | Java | Javascript | Kotlin | Ocaml | PHP | Pascal | Perl | Python | Ruby | Rust | Scala | AVG. |
|---|---|---|---|---|---|---|---|---|---|---|---|---|---|---|---|---|---|---|---|
| Code-Code ($\alpha$) | Acc@$k$ | 56.43 | 56.05 | 39.96 | 62.82 | 66.30 | 56.71 | 49.30 | 69.63 | 63.42 | 58.44 | 64.80 | 52.71 | 56.38 | 55.92 | 61.38 | 58.10 | 66.69 | 58.53 |
| Code-Code ($\gamma$) | Acc@$k$ | 68.66 | 74.50 | 70.49 | 17.35 | 62.62 | 60.03 | 74.71 | 50.70 | 52.06 | 33.72 | 49.88 | 65.35 | 40.50 | 68.33 | 61.71 | 48.58 | 59.76 | 56.41 |
| NL-Code | Acc@$k$ | 82.28 | 89.99 | 83.81 | 68.98 | 90.26 | 81.68 | 84.72 | 85.33 | 84.74 | 85.45 | 80.71 | 82.21 | 81.33 | 84.57 | 87.17 | 82.23 | 89.71 | 83.83 |

performance in *APR*. For *Code Translation*, we observe that Kotlin and Go can be successfully translated to most of the other languages, while C++ and Python are the best languages to translate to.

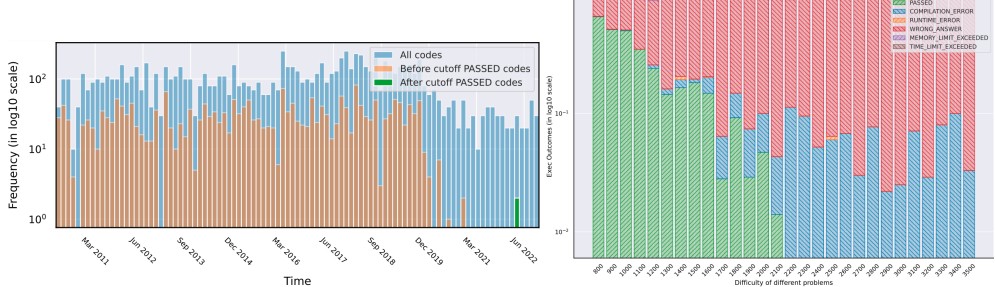

FIGURE 4 − Left: ChatGPT's performance on C++ over time. After the knowledge cutoff (Sep, 2021), the performance is notably poor. Right: distribution of passed solutions (C++) across different difficulty levels.

Table 4 reports results on the two retrieval tasks: *Code-Code* and *NL-Code*. For *Code-Code* retrieval, we computed the top-$k$ retrieval accuracy for each of the 17 query languages from all 17 different retrieval corpora. The summarized results from a $17 \times 17$ matrix (Appendix E.6-Figure 12 of supp. material) are provided in Table 4, where each row represents a query language and each column represents a corpus language. The column-wise and row-wise averages are denoted as ($\alpha$) and ($\gamma$), respectively. For *Code-Code ($\alpha$)*, there is a degradation of performance for languages with large corpus such as C++, Python. In the case of *Code-Code ($\gamma$)*, languages with limited training data in XCODEEVAL , such as *D*, *Ocaml*, Rust performed poorly as query languages. For *NL-Code*, performance is good across all languages except for *D*. We suspect that the limited availability of resources for *D* in both *The Stack* (Kocetkov et al., 2022) dataset (training corpora of StarEncoder) and our XCODEEVAL dataset could account for this discrepancy. Also, the presence of more hard negative candidates (i.e., very similar to a correct code) makes it a challenging task to identify similar codes. We provide more results on the retrieval outcomes in the supplementary material (Appendix E.6).

## 3.2 ANALYSIS

**Knowledge cutoff** XCODEEVAL contains problems that appeared in `codeforces.com` for the timeframe: `Feb 19, 2010 - Nov 21, 2022` (in supp. material Appendix C-Figure 7 shows the distribution over time). Since we have a complete footprint of release dates for each of the problems, it enables us to identify data leakage in LLMs that have public knowledge cutoff dates. Figure 4 (left) presents a potential data contamination for ChatGPT. Though OpenAI (2023) (Table 9) reported no data contamination on `codeforces.com`, datasets like our XCODEEVAL could empower researchers to conduct insightful analysis and perform an investigation on such serious questions. "It should be noted that XCODEEVAL can only analyze data contamination if there is a good amount of problems that appear after the knowledge cut-off date of the evaluated LLM. For a more interpretable evaluation, we invite LLM builders to disclose their knowledge cut-off dates.

**Impact of temperature parameter** Although proved to be crucial (Chen et al., 2021; Austin et al., 2021), previous studies have not extensively examined the impact of the sampling temperature parameter on code executability. To address this gap, we conduct an investigation in which we assessed each sample for *Program Synthesis* at 20 different temperatures in the range $0.0 - 2.0$. Figure 5 (left) presents the overall distribution of *execution outcomes* for various languages, encompassing all the samples generated at different temperatures, while Figure 5 (right) displays a distribution of `PASSED` solutions at different temperatures. As the temperature increases, the likelihood of achieving code

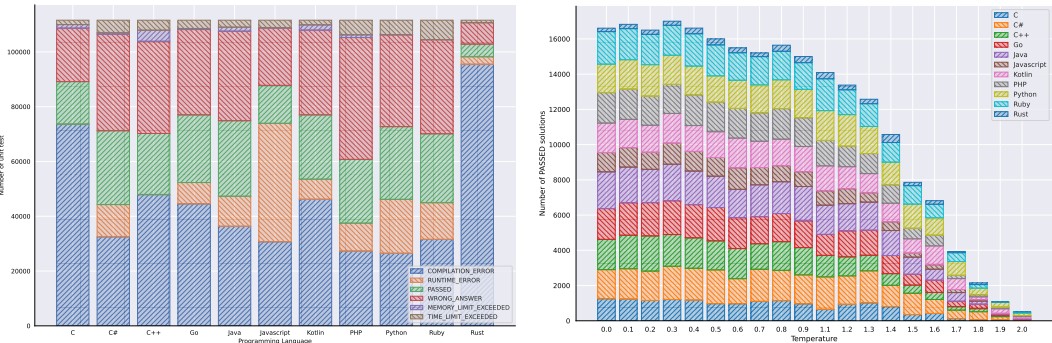

FIGURE 5 – Left: execution outcome for different languages, the solutions are evaluated with `ExecEval` against our unit tests; Right: passed solutions at different temperatures. Both evaluations are done at 20 different temperature values.

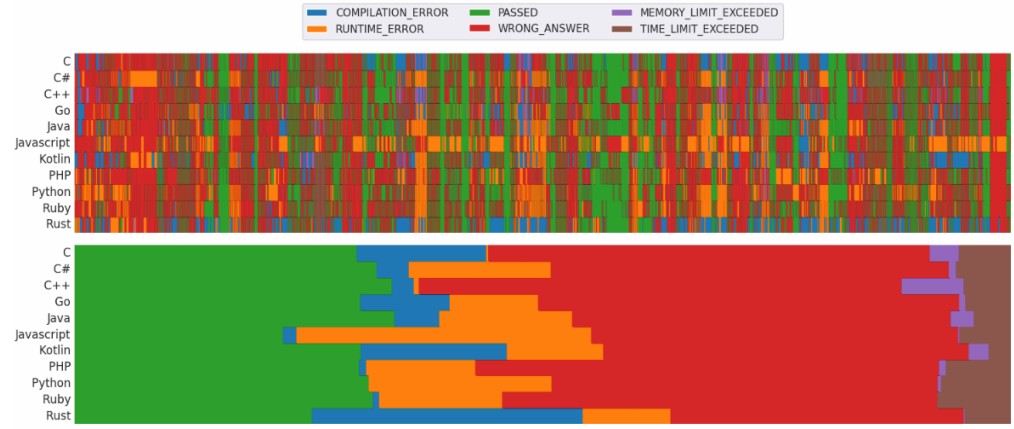

FIGURE 6 – Top: The reasoning spectrum of ***gpt-3.5-turbo-0301***, X-axis represents the unit tests and the color represents the corresponding test outcomes for different languages in the Y-axis; Bottom: The unit tests are grouped together from the reasoning spectrum to get an overall idea of the performance of *execution outcomes*. All the evaluations are done at temperature 0.32 and $n = 10$.

executability decreases. We identify the most successful `PASSED` tests at the temperature of 0.32. Figure 3 (right) presents a comparison of code executability across different languages. For each of the unit test cases, we test it with 20 different temperature values and finally select the temperature with highest `PASSED` execution. We implemented this approach exclusively for a program synthesis task, utilizing this optimal temperature as a pseudo signal for the best parameter for the remaining tasks. While this incurred a significant budget, it is worth noting that with a larger budget, employing optimal parameter search for all tasks and optimal variation would be ideal. In our evaluation, We see that Rust has overall low code executability. On the other hand, interpretable languages like Python, PHP, and Javascript have high executability.

**Difficulty analysis** Figure 4 (right) shows the distribution of `PASSED` problems written in *C++* for different difficulty levels. [5] We see a sharp decrease in performance as the problems become harder.

**Reasoning capability** Figure 6 shows a ***reasoning spectrum*** of ChatGPT on *Program Synthesis*. A program can be considered a reasoning path to produce an output for an input of a unit test. We define *reasoning spectrum* as the range or continuum of different reasoning approaches or strategies that a program can employ to produce an output for a given input in a test case. The reasoning spectrum encompasses various levels of execution outcomes by the program in between different languages. The same colored vertical line along different languages represents an agreement of execution outcome for the corresponding languages. Given any two languages, when the code compiles successfully for both but one produces `PASSED` and the other produces `WRONG ANSWER`, we can conclude that

---

5. Evaluation done at temperature 0.32, generating 10 samples per problem.

there is a gap between reasoning capability of the two languages. We notice a low agreement in the reasoning spectrum between languages suggesting that a further transfer learning mechanism can be applied to transfer reasoning capability from high-resource to low-resource languages.

Though Figure 6 (top) shows a general ***comparable*** structure of reasoning capability of an LLM for different languages, it does not show the overall performance within each language. By grouping the same execution outcomes together, Figure 6 (bottom) shows exceptional code executability on Python (no compilation error). However, their reasoning capability (# of `PASSED` unit tests) remains fairly comparable with other languages.

## 4 EVALUATION OF PROGRAM SYNTHESIS ON SMALLER MODELS

TABLE 5 – Results on Program Synthesis task on validation split. Starcoderbase-3b is finetuned with program synthesis train dataset and zero shot evaluation done for the CodeLlama models.

| Model | Trained | Metric | C | C# | C++ | Go | Java | Javascript | Kotlin | PHP | Python | Ruby | Rust | Avg |
|---|---|---|---|---|---|---|---|---|---|---|---|---|---|---|
| Starcoderbase-3b | ✓ | pass@5 | 1.90 | 1.99 | 3.45 | 1.60 | 2.36 | 2.73 | 2.30 | 2.48 | 2.52 | 2.33 | 1.13 | 2.25 |
| CodeLlama-7b-Instruct | ✗ | pass@5 | 1.12 | 1.74 | 2.64 | 1.65 | 0.87 | 0 | 0.52 | 1.69 | 2.14 | 0.61 | 0.87 | 1.26 |
| CodeLlama-13b-Instruct | ✗ | pass@5 | 4.57 | 4.29 | 6.4 | 2.69 | 3.29 | 2.72 | 4.01 | 3.97 | 4.97 | 2.88 | 2.10 | 3.81 |

For *Program Synthesis* tasks, we fine-tuned `starcoderbase-3B` (Li et al., 2023) model with our trained data. We also evaluated the `CodeLlama-7b-Instruct-hf` and `CodeLlama-13b-Instruct-hf` models with our evaluation data. A 3B fine-tuned model is better than a 7B instruct model but worse than a 13B instruct model. We observe that training a smaller model with the training data performs well on our task rather than using a general-purpose instruct/chat model. However, large instruct models are better than smaller fine-tuned models. So, the impact of our training dataset varies between different scales. Comparing the results between Table 3 and Table 5 also provides a general idea of how challenging our task is.

## 5 DATA LIMITATIONS

Though the codes are written by a diverse group of experts in a diverse number of languages, data is collected from a single source thus limiting the domain diversity. Besides, there is a clear discrepancy between the resource of different programming languages (see Appendix E-Figure 9 in supp. material) and most of the codes in XCODEEVAL are at the document level and often written in a non-modular way without a doc-string. In Appendix K, we discuss the possibilities of evaluation data leakage.

## 6 ETHICS, POTENTIAL RISKS & DOCUMENTATION

Though the data is collected from **openly available sources**, it has not been humanly audited. We have made our best efforts to use automated tools for identifying and removing codes with sensitive details, resulting in the removal of approximately 2 million samples from our original collection. However, it is important to emphasize that despite our diligent efforts, code can still potentially contain sensitive information and security vulnerabilities, although not something that is not openly available. Additionally, code datasets may reflect biases present in the original codebase or the developers who contributed to it.

xCodeEval documentations in supplementary Appendix H to Appendix L, follow all the necessary guidelines of NeurIPS datasets-track (e.g., datasheets (Gebru et al., 2021), nutrition label (Holland et al., 2020), and data card (Hutchinson et al., 2021)). Our github and huggingface repositories provide two valuable sources of both data access and documentation. To mitigate risks and resolve frequently asked questions, we regularly address queries or issues there.

## 7 CONCLUSION & FUTURE WORK

We have introduced XCODEEVAL , a large-scale multilingual multitask benchmark for code-based large language models. XCODEEVAL features seven different tasks involving code understanding, generation, translation and retrieval in up to 17 programming languages, and it employs an execution-based evaluation protocol. We have also presented `ExecEval`, a multilingual code execution engine that supports all the programming languages in XCODEEVAL . In summary, the combination of XCODEEVAL and `ExecEval` presents a novel framework that offers a fresh perspective for examining and analyzing large language models, facilitating comprehensive and to some extent highly interpretable investigations. We hope that by utilizing the extensive metadata and execution-based evaluation, there is potential for the discovery of new scaling laws and emergent capabilities.

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

APPENDIX

## A FREQUENTLY ASKED QUESTIONS

**Notes on Intended Usage Scenarios**   Considering the recent progress, the primary objective of creating XCODEEVAL is to challenge LLMs, especially the **frontier models**. As such, the tasks proposed could be very challenging for smaller models, even for the binary *code compilation* task. The relatively smaller (per-language) *validation* splits can be used to assess multilingual features and to get an overall picture of multilingual generalization. The large **Test** split is meant to rigorously evaluate specific programming languages and conduct more language-specific, in-depth analysis. We have also included the **Training** split. The intended use of the training split is to include it in the large scale pre-training or SFT mixtures. For both **validation** and **test** evaluation, we recommend using an **Instruct/Chat** Model.

**Where are the Dataset Construction recipes?**   This paper provides a detailed description of our dataset construction. However, to adhere to the page limit, we needed to shorten them in the main paper (Section 1, 2) and move to the supplementary appendix Appendix E for details.

**Where are the documenation, maintenance, support?**   XCODEEVAL documentations in supplementary Sec Appendix H to Appendix M, follow all the necessary guidelines of datasheets (Gebru et al., 2021), nutrition label (Holland et al., 2018), and data card [6]. Following the accountability framework proposed by (Hutchinson et al., 2021), we released the data construction procedures as well as opensource the evaluation framework [7]. Our github [8] and huggingface repositories [9] provide two valuable sources of both data access and additional documentation and implementation details. We regularly address queries or issues there. If any specific documentation is missing, we would be happy to address them right away.

**Is there any automated process used for creating the Gold Labels?**   Please note that all current data (text, code, test-cases, and labels) are human written. More specifically for tag classification tasks, the tags are already provided in codeforces.com. The tags are generally updated after the contest by experienced problem solvers trusted by the codeforces team. In addition to that, the evaluation is done by programming language compilers & interpreters. We put huge effort in developing ExecEval and dockerized it to synchronize the evaluation process.

**On the possibility of data leakage and contamination**   We note that, although we completely separate our validation and test data from training, LLMs might have possible data leakage from pretraining. We find that even identifying data leakage (test data exists or not in the prertraining corpora) is challenging using conventional data search methods due to search cost and complexity (e.g., exact match or token overlapping methods) for (i) long sequence search for libraries (ii) boilerplate code identifying. Apart from that, hashing based searches often suffer from not having properly segmented text.

In this paper, we introduce an approach to address the challenge of leakage-free evaluation, employing a technique rooted in the concept of a **knowledge cut-off** Our finding in Section 3.2 in the main paper shows that the data contamination significantly impacts the model performance and it needs to be interpreted with proper analyses. Another method toward leakage less evaluation could be to have a human written separate evaluation set that is hidden or regularly updated which we are considering in our long term release plans.

**Only generative tasks utilize unit tests. How are classification and retrieval tasks considering executability?**   In our proposed tasks, except for the tag classification task, all tasks consider **online** or **offline** unit test executability. We included a tag classification task since tags were used for the sampling algorithm that we proposed. Other than that, our code compilation and retrieval task takes into account offline code executability, where compilers are applied before releasing datasets to obtain labels. Additionally, in retrieval tasks, we treat passed samples as correctly retrieved samples and generate hard negatives from incorrect code from the same problem. Please note that all the data—including text, code, test cases, and labels—is human-written.

---

6. https://sites.research.google/datacardsplaybook/
7. https://github.com/ntunlp/ExecEval
8. https://github.com/ntunlp/xCodeEval
9. https://huggingface.co/datasets/NTU-NLP-sg/xCodeEval

TABLE 6 – Comparison between XCODEEVAL and other benchmarks. For simplicity, we combine NL-code generation and code completion as Program Synthesis. Compared to others, XCODEEVAL offers the largest suite of training and test data and a more comprehensive set of test cases. Evaluation levels `Global`, `Modular`, and `Local` refer to document, function, and statements level evaluation, respectively.

| Dataset | \|Train\| | \|Test\| | \|La\| | Task Type | Evaluation | Level | Genre |
|---|---|---|---|---|---|---|---|
| Django (Oda et al., 2015) | 16,000 | 1,805 | 1 | Program Synthesis | Lexical | Local | N/A |
| WikiSQL (Zhong et al., 2017) | 56,355 | 15,878 | 1 | SQL Queries | Lexical | Modular | SQL |
| Miceli Barone and Sennrich (2017) | 109,108 | 2,000 | 1 | Synthesis, Summarization | Lexical | Local | Github |
| CoNaLa (Yin et al., 2018) | 2,379 | 500 | 2 | Program Synthesis | Lexical | Local | Stackoverflow: QA |
| CONCODE (Iyer et al., 2018) | 100,000 | 2,000 | 1 | Program Synthesis | Lexical | Modular | Github |
| Android (Parvez et al., 2018) | 26,600 | 3,546 | 1 | Program Synthesis | Lexical | Local | Map oriented, GitHub |
| CodeSearchNet (Husain et al., 2019) | 6,452,446 | 99 | 6 | Plain Text, Retrieval | NDCG | Modular | Github |
| JuICe (Agashe et al., 2019) | 1,518,049 | 1,981 | 1 | Notebook Cell Gen. | Lexical | Local | Prog. assignment |
| TransCoder (Roziere et al., 2020) | 721MB | 1,410 | 3 | Program Translation | Lexical | Modular | Github |
| HumanEval (Chen et al., 2021) | - | 164 | 1 | Program Synthesis | Execution | Modular | Interview Question |
| HumanEval-X(THUDM, 2022) | - | 820 | 9 | Synthesis & Translation | Execution | Modular | Interview Question |
| MBPP (Austin et al., 2021) | - | 974 | 1 | Program Synthesis | Execution | Modular | Interview Question |
| CodeXGLUE (Lu et al., 2021) | 2,840,000 | 759,000 | 9 | 10 Tasks | Lexical | Local | N/A |
| AVATAR (Ahmad et al., 2021b) | 5,937 | 1,693 | 2 | Program Translation | Lexical | Global | Problem Solving |
| TFix (Berabi et al., 2021) | 84,846 | 10,504 | 1 | Program Repair | Lexical | Local | Github |
| CCSD (Liu et al., 2021) | 84,316 | 6,533 | 1 | Program Summarization | Lexical | Modular | Linux Kernel |
| TL-CodeSum (Hu et al., 2018) | 55,766 | 6,971 | 1 | Program Summarization | Lexical | Modular | Github |
| CodeNet (Puri et al., 2021) | 8,906,769 | 2,783,365 | 55 | Classification, similarity | Lexical | Global | Problem Solving |
| TransCoder-ST (Roziere et al., 2021) | 333,542 | 103,488 | 3 | Program Translation | Execution | Modular | Github |
| DSP (Chandel et al., 2022) | - | 1,119 | 1 | Notebook Cell Gen. | Execution | Local | Math and Data Science |
| MTPB (Nijkamp et al., 2022) | - | 115 | 1 | Multi-turn Code Gen. | Execution | Local | Problem Solving |
| Exe-DS (Huang et al., 2022) | 119,266 | 534 | 1 | Notebook Cell Gen. | Execution | Local | Data Science |
| DS-1000 (Lai et al., 2022) | - | 1,000 | 1 | Notebook Cell Gen. | Execution | Local | Data Science |
| MoCoNaLa (Wang et al., 2022a) | - | 896 | 1 | Program Synthesis | Lexical | Local | StackOverflow |
| ARCADE (Yin et al., 2022) | - | 1,082 | 1 | Notebook Cell Gen. | Lexical | Local | Data Science |
| ODEX (Wang et al., 2022b) | - | 945 | 1 | Program Synthesis | Execution | Local | StackOverflow |
| MBXP (Athiwaratkun et al., 2022) | - | 13,877 | 10 | Program Synthesis | Execution | Modular | Interview Question |
| XLCoST(Zhu et al., 2022) | 496,333 | 45,394 | 7 | 10 Task | Lexical | Local, Global | GitHub |
| DeepFix (Gupta et al., 2017) | 37,000 | 7,000 | 1 | Program Repair | Ececution | Global | Compile Error, Students |
| Defects4J (Just et al., 2014) | - | 835 | 1 | Program Repair | Execution | Local, Global | N/A |
| APPS (Hendrycks et al., 2021) | 5,000 | 5,000 | 1 | Program Synthesis | Execution | Global | Interview Question |
| CodeContests (Li et al., 2022) | 4,432,447 | 32,181 | 3 | Program Synthesis | Execution | Global | Problem Solving |
| CoderEval (Yu et al., 2023) | - | 460 | 2 | Program Synthesis | Execution | Modular, Global | GitHub |
| Humanevalpack (Muennighoff et al., 2023) | - | 6×164 | 6 | Program Synthesis | Execution | Modular | Interview Question |
| BioCoder (Tang et al., 2023) | - | 2,522 | 2 | Program Synthesis | Execution | Modular, Global | Github |
| CodeApex (Fu et al., 2023) | - | 706 | 1 | 3 tasks | Execution | Modular | Online Judge platform |
| **XCODEEVAL** (ours) | 19,915,150 | 159,464 | 17 | 7 Tasks, see Table 8 | Execution | Global | Problem Solving |

# B   RELATED WORK

Following NLP (Devlin et al., 2019; Radford et al., 2018; Raffel et al., 2020), transformer-based pre-trained LLMs have shown significant success in code, both in understanding and generation. Table 6 shows a detailed comparison between different programming language-related datasets.

**Code Understanding**   Lu et al. (2021) propose a benchmark CodeXGLUE, which comprises three widely-used code understanding tasks, *defect detection*, *clone detection*, and *code search*. Zhou et al. (2019) treat *defect detection* as a binary classification task. They propose a model called *Devign* which they evaluate on four large open-source C projects. Additionally, Russell et al. (2018) leverage open-source C/CPP repositories to support function-level vulnerability detection. To further understand code semantics, Svajlenko et al. (2014) propose a benchmark BigCloneBench to measure the similarity between code pairs to predict whether they have the same functionality (i.e., clone detection); BigCloneBench was collected from open-source Java repositories with manual validation. Arguably, code defect and clone detection might not be appropriate for fully evaluating models' ability in understanding code semantics (Wang et al., 2021; Guo et al., 2022). Moreover, they only support a few programming languages. *Code search* on the other hand considers semantic relevance for both code-to-code and text-to-code. They are formulated to retrieve semantically similar codes given a query code (Lu et al., 2021) or code description (Husain et al., 2019). The existing code search benchmarks like CodeSearchNet (Husain et al., 2019) only select the first documentation as the text query to search corresponding functions. Recently, Fu et al. (2023) introduce CodeApex, a bilingual benchmark to evaluate the language models on three different tasks consisting of *programming comprehension, code generation, code correction*. Among its tasks, programming comprehension examines the ability to understand code from various aspects, such as the syntax's mastery, code execution flow, and executing algorithms. Nonetheless, this dataset only covers one programming language, which is in contrast to our work.

**Code Generation**   Code generation has grown in popularity as many pre-trained LLMs have achieved remarkable performances in these tasks like decoder-only models (Chen et al., 2021; Izadi et al., 2022; Nijkamp et al., 2022) and encoder-decoder models (Wang et al., 2021; Guo et al., 2022;

Ahmad et al., 2021a) Notably, PaLM (Chowdhery et al., 2022) and AlphaCode (Li et al., 2022) outperform average human participant in competition-level coding. Thus, researchers attempt to build increasingly difficult and factual code generation tasks. These tasks can be classified as code-to-code generation and text-to-code generation.

As for code-to-code generation tasks like automatic program repair (APR) (Tufano et al., 2019) and code translation (Lu et al., 2021), the metric-based automatic evaluation measures like BLEU (Papineni et al., 2002), CodeBLEU (Ren et al., 2020), and exact match scores are sub-optimal for evaluating the quality of a generated code. To improve the reliability and feasibility for code generation evaluation, Berabi et al. (2021) create a large-scale JavaScript patch repair dataset from GitHub commits, where 52 error types are detected by a static analyzer ESLint [10]. They further drive efforts in enhancing program repair evaluation by providing an error removal metric to take various form of error fixes into consideration. To address the nature of code semantic and syntactic evaluation, execution-based evaluation with comprehensive test suites has a growing demand. A popular Java APR benchmark Defects4J (Just et al., 2014) takes the number of correct fixes into account, where a correct fix should pass all test cases and provide a desired functionality. Nevertheless, Defects4J does not possess a cohesive training corpus. A common strategy to address this limitation is to construct the training dataset using GitHub's publicly available repositories, and relying on bug-specific commit messages (Zhu et al., 2021). However, this heuristic-based approach includes bug-irrelevant commits and unrelated code pairs, which can significantly affect the the quality of collected training dataset (Xia and Zhang, 2022).

For text-to-code, the widely used dataset CONCODE (Iyer et al., 2018) consists of a large collection of natural language (NL) comments and Java code snippets. Specifically, this dataset is constructed by scraping code snippets from open-domain Java GitHub repositories and utilizing heuristics to extract NL comments from Javadoc.

By following a similar approach, JuICe (Agashe et al., 2019) collects publicly available Jupyter notebooks from GitHub, and CoNaLa (Yin et al., 2018) collects Python and Java codes with NL comments from StackOverflow posts. Besides, they attempt to improve the quality with professional annotators. In addition, MoCoNaLa (Wang et al., 2022a) extends CoNaLa to support more natural languages. Despite their coverage, the general lexical-based evaluation metrics are insufficient to measure the correctness of generated codes. To alleviate this limitation, ODEX (Wang et al., 2022b) provides execution-based evaluation via human-written test cases of diversified Python libraries. This execution-based paradigm has been widely applied to evaluate benchmarks in Data Science domain like DSP (Chandel et al., 2022), DS-1000 (Lai et al., 2022) and Exe-DS (Huang et al., 2022) as well as general code generation benchmarks in single-language settings such as HumanEval (Chen et al., 2021), MBPP (Austin et al., 2021), and APPS (Hendrycks et al., 2021). Apart from HumanEval, CoderEval (Yu et al., 2023) further leverages the contextual information and achieves a full spectrum of test coverage with additional manually crafted tests, providing 100% test coverage. To improve the diversity of code generation tasks, Fu et al. (2023) propose a bilingual code evaluation benchmark CodeApex to support both English-to-Code and Chinese-to-Code generation tasks. As for more particular multi-turn MTPB (Nijkamp et al., 2022), multi-language CodeContests (Li et al., 2022), and domain specific BioCoder (Tang et al., 2023) benchmarks, they all leverage test cases, and exploit code execution for better evaluation.

## C  ALGORITHM FOR INITIAL VALIDATION AND TEST SPLIT CREATION

To make sure we do not have train and (validation, test) overlap, at first we divide the set of problems into two sets. In one set we keep all the problems for which we do not have a complete set of unit tests. In another set, we keep the problems where we have a complete set of unit tests that ensures the correctness of the solution of the problem. We use the first set for training and the latter set for validation and test data. Figure 7 shows the chronological distribution of our training, validation, and test data. After selecting validation and test problem sets, we have thousands of solutions for each of the problems. But these problems are not divided into validation and test splits. As a heuristic, we can consider the tag distribution as the domain of the problem. To ensure that we have proper domain coverage we employ Algorithm 1. Algorithm 1 ensures that validation and test sets contain the same tag sets as the training set. In addition to that, it also selects the best possible splitting point based on

---

10. https://eslint.org

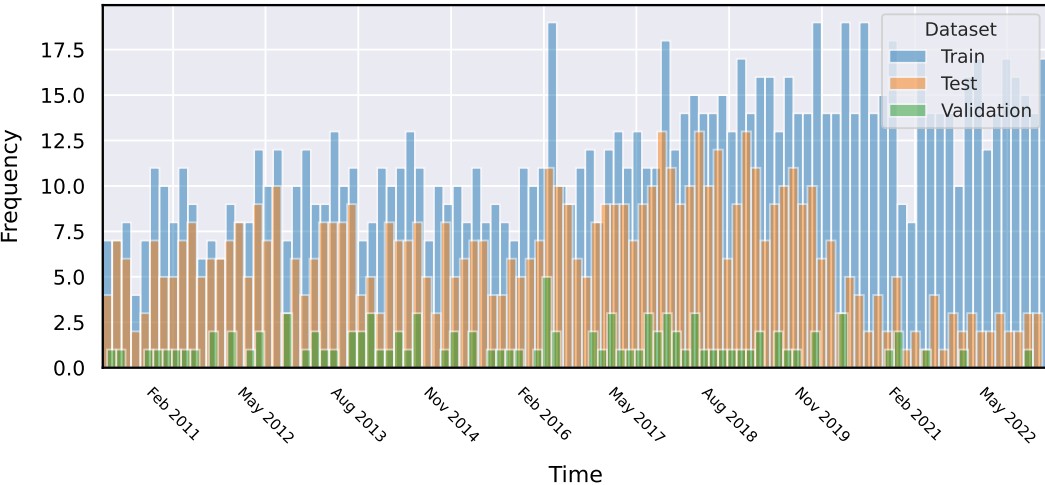

FIGURE 7 – The chronological order of the problems' online appearance for the first time

the geometric mean. However algorithm 1 provides only a splitting point for hundreds of thousands of validation and test data points. To reduce the redundant data based on different levels of conditions, we formulate the problem of selecting data points to a linear programming problem (more on this in Section 2.1).

---

**Algorithm 1** Validation and Test Split Creation

---

**Input:** A held-out dataset $\mathcal{D}_{\text{ho}}$, a fraction value $\gamma$ where $0 \leq \gamma \leq 1$, an integer $N$ indicating number of seeds.
**Output:** $\mathcal{D}_{\text{valid}}$, $\mathcal{D}_{\text{test}}$ spits
**Initialize:** $count = 0, bestScore = \gamma + 1$
**while** $count < N$ **do**
   $seed = getSeed()$
   Shuffle $\mathcal{D}_{\text{ho}}$
   $\mathcal{D}_{\text{valid}} = \mathcal{D}_{\text{ho}}[0 : |\mathcal{D}_{\text{ho}}| \times \gamma]$
   $\mathcal{T}_{\text{valid}} = $ set of tags in $\mathcal{D}_{\text{valid}}$
   $\mathcal{D}_{\text{test}} = \mathcal{D}_{\text{ho}}[|\mathcal{D}_{\text{ho}}| \times \gamma : |\mathcal{D}_{\text{ho}}|]$
   $\mathcal{T}_{\text{test}} = $ set of tags in $\mathcal{D}_{\text{test}}$
   **if** $\mathcal{T}_{\text{valid}} \neq \mathcal{T}_{\text{test}}$ **then**
      continue
   **end if**
   **for all** $T$ in $\mathcal{T}_{\text{valid}}$ **do**
      $\gamma_T = \frac{\#\text{samples in } \mathcal{D}_{\text{valid}} \text{ with tag } T}{\#\text{samples in } \mathcal{D}_{\text{test}} \text{ with tag } T}$
   **end for**
   $\mu = geoMean(\{\gamma_T\}_{T \in \mathcal{T}_{\text{valid}}})$
   **if** $|\gamma - bestScore| > |\gamma - \mu|$ **then**
      $bestScore = \mu$
      save current split $\{\mathcal{D}_{\text{valid}}, \mathcal{D}_{\text{test}}\}$
      $count = count + 1$
   **end if**
**end while**

---

## D  HYPER PARAMETER TUNING FOR CIRCULATION PROBLEM

Let $M$ be the number of samples we want to select for any set of submissions. We call any $(m_p, m_t, x_p, x_t)$ a valid tuple if the flow network has a feasible flow for the circulation problem defined in eq. in 2.1. Let $d = \lfloor (\sum_{i=1}^{N} f(s, P_i) - M)^2 / \Delta \rfloor$, representing the squared difference between samples we want and the samples selected for the flow and $\Delta$ reduces the resolution in which

we look for differences. Here $d$ defines a boundary from $M$ where we allow choosing an expected solution with $m_p$, $m_t$, $x_p$, and $x_t$. Finally, the lexicographical ordering $(-d, m_t, -x_t, -x_p, m_p)$ is used to find the largest element in the collection of valid tuples which always exist if we limit our search space to a finite set. The largest element in this ordering depicts the nearest (close to $M$) selection of samples that maximizes the minimum number of samples per tag $m_t$. When there are many solutions with the same $(-d, m_t)$, we prioritize reducing the maximum number of samples per tag, $x_t$. Similarly, we prioritize $x_p$ and $m_p$ as defined in the lexicographical ordering.

### D.1    SEARCH TECHNIQUES

1. It was manually checked that $(m_p, m_t, x_p, x_t) = (1, 1, 1000, 1000)$ is a valid tuple for any set of submissions that were processed and $\Delta = 1000$ was chosen.

2. In Tag classification task (Section E.1) and Code compilation task (Section E.2), $M$ is 2000, 10000 for any language for validation, test split respectively. For Code translation (Section E.4) $M$ was 400, 2000 for the same.

3. Search largest tuple $(-d_1, m_{t_1}, -x_{t_1}, -x_{p_1}, m_{p_1})$ where $m_{t_1} \in \{1, 6, 11, \cdots, 496\}$, $m_{p_1} \in \{1, 2, 3, \cdots, 19\}$ and $x_{p_1} = x_{t_1} = 1000$. Since $(m_p, m_t, x_p, x_t) = (1, 1, 1000, 1000)$ is a valid solution, hence the set of valid tuples is nonempty. Let $f_1$ be the flow for the flow network defined for $m_{t_1}, -x_{t_1}, -x_{p_1}, m_{p_1}$. Let $f_{P_1} = \max_{1 \le i \le N} f_1(s, P_i)$, $f_{T_1} = \max_{1 \le k \le K} f_1(T_k, t)$ be the maximum flow through edges from $s$ to $P_i$, and same through edges from $T_k$ to $t$.

4. Now again search the largest tuple $(-d_2, m_{t_2}, -x_{t_2}, -x_{p_2}, m_{p_2})$ where $m_{t_2} \in \{m_{t_1}, m_{t_1} + 1, \cdots, m_{t_1} + 49\}$, $x_{t_2} \in \{f_{T_1} - 100, f_{T_1} - 80, \cdots, f_{T_1}\}$, $x_{p_2} \in \{f_{P_1} - 5, f_{P_1} - 4, \cdots, f_{P_1}\}$, $m_{p_2} \in \{m_{p_1}, m_{p_1} + 1\}$. Since $m_{t_1}, f_{T_1}, m_{p_1}, f_{P_1}$ is included a solution is found in this step too. Define $f_{P_2}, f_{T_2}$ similar to previous step.

5. Finally search the largest tuple $(-d_3, m_{t_3}, -x_{t_3}, -x_{p_3}, m_{p_3})$ where $m_{t_3} = m_{t_2}, x_{t_3} \in \{f_{T_2} - 100, f_{T_2} - 99, \cdots, f_{T_2}\}, x_{p_3} = x_{p_2}, m_{p_3} = m_{p_2}$.

While it is not an exhaustive search, we prioritize minimizing $x_t - m_t$ over $x_p - m_p$.

### D.2    RESULTS

We compared the performance of data selection using circulation problem technique with randomly selecting equal number of samples for validation and test sets of all languages and measured the skew $\tilde{\mu}_3$, and standard deviation $\sigma$ of the distribution of tags in the selected data. Here lower value of $|\tilde{\mu}_3|$ means more symmetric distribution. On the other hand, a lower value of $\sigma$ represents that the number of samples in each tag are closer to the mean.

## E    TASKS CONSTRUCTION PROCESS

### E.1    TAG CLASSIFICATION

We formulate the task as a multi-label classification problem in two settings: Code-to-Tag (Code2Tag) and Problem Description-and-Code to Tag (DesCode2Tag). In Code2Tag, given a code $C$ in any language, the task is to predict the corresponding tag set $\mathbb{T}$. In DesCode2Tag, the natural language problem description is also given as input in addition to the code. The performance difference between *Code2Tag* and *DescCode2Tag* settings can suggest if the problem description can help models identify the problem tags (i.e., the type of solution needed)..

For these tasks, the split for validation and test is done with a ratio of $1 : 5$ (i.e., $\gamma = 0.2$) using Algorithm 1. To get the final $\mathcal{D}_{\text{valid}}$ and $\mathcal{D}_{\text{test}}$ with a feasible number of samples, we use flow network-based data selection approach with the details of hyper-parameter settings presented in Section 2.1.

The distribution of the samples according to the tags is presented in Fig 8.

We further propose a language-specific tag classification task, in which each programming language has its own *Code2Tag* and *DesCode2Tag* settings.

TABLE 7 – Comparison of skew and standard deviation of tags using circulation problem technique and random data selection (lower value is better).

| Language | Skew, $\tilde{\mu}_3$ | | | | Std. deviation, $\sigma$ | | | |
| | Validation | | Test | | Validation | | Test | |
| | Random | Circ. | Random | Circ. | Random | Circ. | Random | Circ. |
|---|---|---|---|---|---|---|---|---|
| **Tag Classification** | | | | | | | | |
| C | 2.778 | 2.499 | 2.848 | 2.440 | 249.161 | 213.849 | 880.881 | 772.549 |
| C++ | 2.405 | 1.873 | 2.315 | 1.655 | 233.530 | 157.889 | 1154.538 | 751.023 |
| Python | 2.731 | 2.365 | 2.689 | 2.173 | 265.193 | 240.248 | 1125.133 | 992.904 |
| Java | 2.652 | 1.990 | 2.545 | 2.050 | 258.587 | 207.881 | 1175.790 | 972.703 |
| C# | 3.066 | 2.598 | 2.971 | 2.506 | 314.219 | 291.813 | 846.426 | 760.069 |
| **Code Translation** | | | | | | | | |
| C | 2.744 | 2.455 | 2.941 | 2.332 | 117.298 | 99.261 | 267.214 | 215.881 |
| C++ | 2.424 | 2.112 | 2.287 | 1.565 | 131.632 | 120.979 | 243.100 | 150.498 |
| Python | 2.533 | 2.379 | 2.635 | 2.294 | 123.710 | 110.076 | 271.219 | 237.179 |
| Java | 2.558 | 2.208 | 2.605 | 1.827 | 134.314 | 114.840 | 259.510 | 193.211 |
| C# | 3.147 | 2.532 | 2.943 | 2.395 | 103.838 | 96.747 | 250.049 | 220.615 |
| PHP | 2.506 | 2.744 | 2.520 | 2.730 | 59.321 | 59.877 | 270.582 | 278.530 |
| Rust | 2.520 | 2.393 | 2.534 | 2.311 | 59.269 | 60.253 | 269.352 | 264.507 |
| Go | 2.807 | 2.359 | 2.676 | 2.424 | 72.415 | 66.666 | 266.565 | 254.986 |
| Javascript | 2.611 | 2.611 | 2.473 | 2.473 | 64.090 | 64.090 | 246.483 | 246.483 |
| Ruby | 2.875 | 2.686 | 2.968 | 2.762 | 74.153 | 70.760 | 280.000 | 271.539 |
| Kotlin | 2.865 | 2.576 | 3.108 | 2.534 | 59.765 | 56.114 | 266.430 | 257.155 |

TABLE 8 – Size of the datasets for each task and the evaluation metrics. For *Program Synthesis* train data `{problem description, solution}` comes from **7514** problems of 11-17 languages where the input for validation and test data is only natural language text (problem description) independent of programming languages. For all other tasks, validation and test samples are reported for a total number of languages.

| Task Type | Task | \|Lang\| | \|Train\| | \|Validation\| | \|Test\| | Metric |
|---|---|---|---|---|---|---|
| Classification | Tag Classification | 11 | 5,494,008 | 18,696 | 74,733 | macro-f1 |
| | Code Compilation | 11 | 19,915,150 | 6,394 | 30,388 | accuracy |
| Generative | Program Synthesis | 11 | 5,538,841 | 106 | 952 | pass@k |
| | Code Translation | 11 | 5,538,841 | 7,034 | 20,356 | pass@k |
| | Automatic Program Repair | 11 | 4,672,070 | 5,068 | 17,699 | pass@k |
| Retrieval | Code-Code Retrieval | 17 | 45,270 | 2,335 | 9,508 | Acc@k |
| | NL-Code Retrieval | 17 | 55,924 | 2,780 | 11,157 | |

## E.2  CODE COMPILATION

Given a code $C$ in a language $L$ and its compiler or interpreter version $B$, the *code compilation* task is to decide whether the code compiles or not. The validation and test splits are created using a modified version of Algorithm 1 that balances the partition based on the compilation outcome of the code instead of the tags of the problem that the code belongs to. We use a ratio $\gamma$ of $1 : 5$. Then a simplified version of the circulation problem is used to prevent too many codes coming from a single problem, and also to ensure a balanced output distribution. The details of hyper-parameter settings of

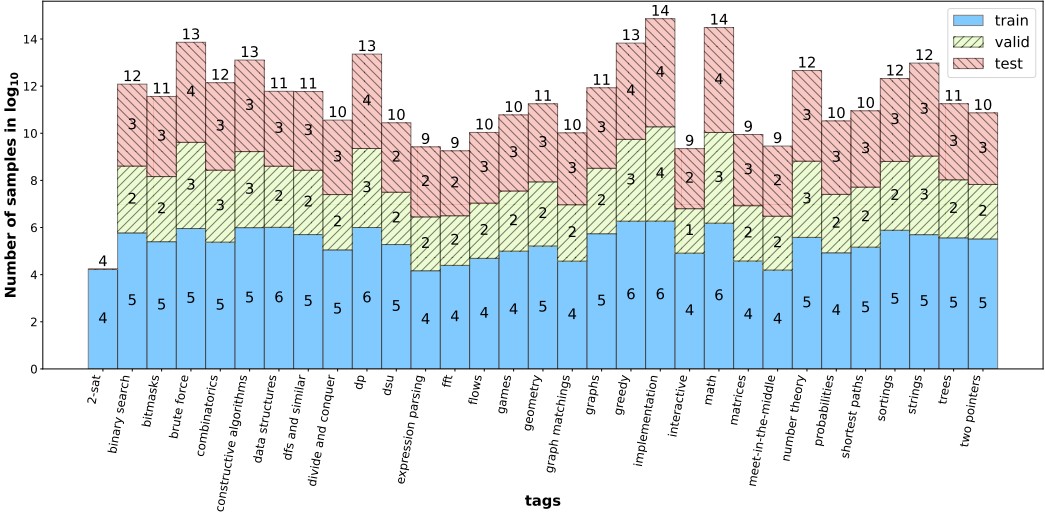

FIGURE 8 – Tag distribution in XCODEEVAL. In XCODEEVAL, often multiple tags are assigned to the same problem as there are usually many different ways to solve a problem or it may require a combination of different approaches.

the circulation problem technique are presented in Section 2.1. In the flow network construction, tags $\{T_k\} = \{true, false\}$ as *true* if the code compiles or not. Furthermore true and false examples are present in equal numbers in both validation and test dataset.

We propose three generative tasks which require a global understanding of programming languages. For the evaluation of generative tasks, we follow execution-based evaluation instead of lexical similarity. All the generative tasks are evaluated using `ExecEval` execution engine. We provide complete unit tests for all problems in the validation and test dataset which also satisfy the conditions of the input-output description of the problem.

### E.3    PROGRAM SYNTHESIS

Given a problem described in natural language, program synthesis task is to write a program that solves the problem. We can express each sample in the dataset as a tuple $(C, P, l, L)$, where $C$ denotes a solution code written in a programming language $L$ for the problem $P$, and $l$ denotes the compiler/interpreter version of the code. All code samples in our dataset are unique and marked as a correct solution (`PASSED` outcome) to the problem. The validation and test splits are created from the heldout problems using Algorithm 1 with a ratio ($\gamma$) of $1 : 9$. The generated code is judged based on executions on the unit tests.

### E.4    CODE TRANSLATION

Each sample in the code translation data can be expressed as a tuple $(\mathcal{C}, P, l, L)$, where $\mathcal{C}$ denotes a set of solution codes in a programming language $L$ for the problem $P$, and $l$ denotes the compiler/interpreter version of the code. All codes in set $\mathcal{C}$ are unique and guaranteed to be marked as a correct (`PASSED` outcome) solution to the problem by the compiler/interpreter.

The validation and test splits are created from the held-out problems using Algorithm 1 with a ratio ($\gamma$) of $1 : 5$, and employ the data selection method with flow network (Sec. 2.1) to have a practical evaluation setup while ensuring a balanced distribution over problems and tags. Figure 9 shows the distribution of the machine translation tasks. Since code translation considers all possible directions of translation between languages, in addition, to train, validation, and test split, we also provide a small validation split.

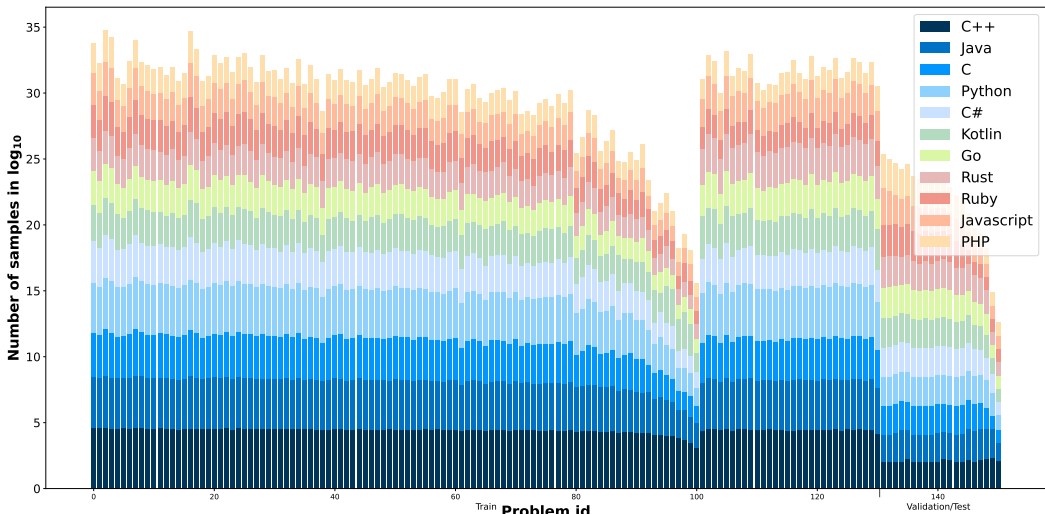

FIGURE 9 – Distribution of samples across all problems in the train, validation, test splits for all languages in the machine translation task.

### E.5 AUTOMATIC PROGRAM REPAIR (APR)

We consider APR as a task to synthesize a fix for a detected program bug. We create a bug-fix pair by matching a buggy code (1-5 execution outcome in Sec. 2.2) with a `PASSED` solution. Given a bug-specific defect, the objective of this task is to generate a correct fix that passes all the unit tests.

Let $\mathbb{C} = \{C_1, \dots, C_m\}$ be the set of programs submitted by a participant in a chronological order in order to solve a specific problem $P$. Some of these submissions can be 'buggy', while some can be `PASSED`. We create the 'bug-fix' pairs from $\mathbb{C}$ as follows.

1. We iterate over $\mathbb{C}$ and mark the `PASSED` ones as 'fixed'. Let $C_j^*$ is one such case.
2. For each buggy submission that was made before $C_j^*$, we measure its lexical similarity with $C_j^*$ and select the one (say $C_k$ where $k < j$) with the highest similarity score to pair it with $C_j^*$ and form a bug-fix pair $(C_k, C_j^*)$. We use `difflib`[11] to measure the similarity.
3. With each bug-fix pair $(C_k, C_j^*)$, we also include the corresponding problem description $P$ and execution outcome $V_k$ (Section 2.2) of $C_k$.
4. The tuple $(C_k, C_j^*, P, V_k)$ represents a sample in our APR task.

We repeat this process for each participant and problem to create the final APR dataset. As reported in Table 8, it comprises more than 5M practical bug-fix pairs and supports 11 programming languages. For data selection in APR, we considered execution outcome (section 2.2) as *tags* in the network flow construction (section 2.1).

Due to the large input specification of the APR task, sometimes the input sequence length becomes too large. However, we have not compromised the benchmarks by selecting only small sequence length samples but rather keep them as challenging tasks for the language models. Figure 10 shows the length distribution of validation and test input sequence.

### E.6 CODE RETRIEVAL

Code retrieval tasks typically aim to measure the mere semantic relatedness between a natural language (NL) query and a programming language (PL) code. However, a code that is relevant, can still be buggy and thus be misleading (see an example in Figure 11). In view of this, we propose two new and more challenging retrieval tasks in our benchmark, which require a deeper understanding of the NL query and code. In particular, we propose NL-Code and Code-Code retrieval tasks that

---

11. https://docs.python.org/3/library/difflib.html

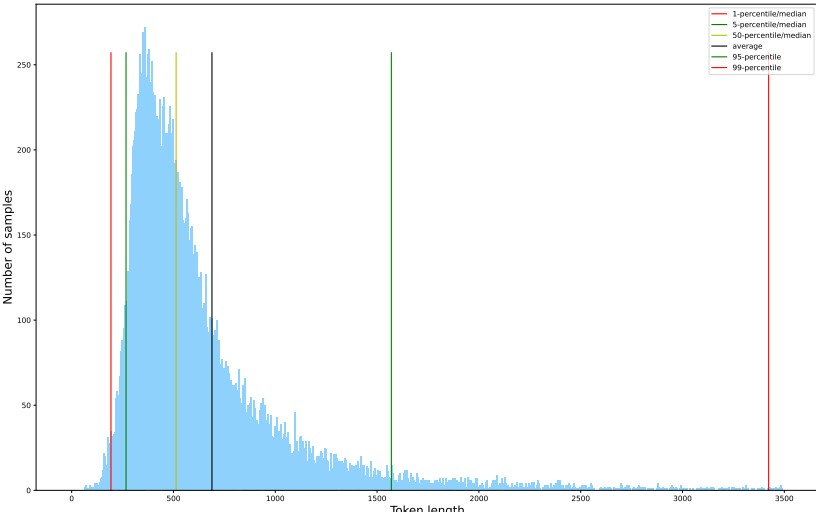

FIGURE 10 – Distribution of sequence length of tokenized (`bigscience/bloom`) samples in the validation and test splits for all languages in the APR task (Section E.5). Each sample contains a buggy code with it's problem description.

involve identifying a *correct code* from a large pool of candidates containing similar codes. In both tasks, for each programming language, we aggregate all the submitted codes and their test cases to create a retrieval corpus and a testbed for evaluating their correctness against test cases. Figure 9 gives a detailed statistics of our retrieval tasks. The datasets for the subtasks and the evalaution schema are discussed below.

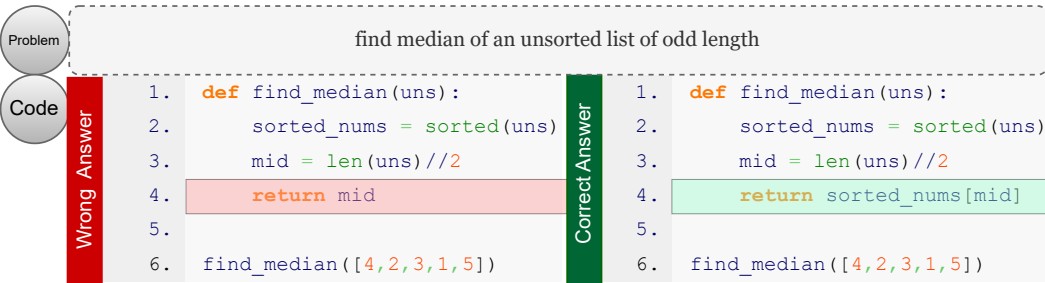

FIGURE 11 – A code retrieval example. The candidate code on the left has a bug highlighted in red and that on the right has a fix highlighted in green. Both our proposed NL-Code and Code-Code retrieval tasks ensure differentiating between them and pose a more challenging task that aims to comprehend both semantic and logical relatedness.

**NL-Code Retrieval**   This task involves matching an NL problem description to the most relevant and correct code from a pool of candidates. An example of an NL description and its corresponding codes are shown in Figure 1. To gather data for this task, we only use instances where the NL description is valid and there is at least one correct solution code (i.e., with execution outcome `PASSED`). For an NL problem description, we consider all the correct solutions as positive examples and all the wrong (or buggy) solutions as negative examples.

**Code-Code Retrieval**   Given an input code (as a query), this task involves finding similar and logically equivalent code (i.e., passes the same set of test cases) from a collection of candidates. We ensure that the query code solves a specific problem (i.e., a correct solution without any detected bugs) and evaluate whether the retrieved candidate also solves the same problem or not. To collect data for this task, we only consider the programming problems which have at least *two* correct code solutions that pass all the corresponding test cases (i.e., with execution outcome `PASSED`).

TABLE 9 – Retrieval subtasks statistics. |Size| denotes the number of instances. For each train/dev instance, we provide multiple positive and negative examples, and |Pos| and |Neg| refer to the total number of positive and negative annotations.

| Lang | Subtask | Train | | | Dev | | | Test | Retrieval |
| | | |Size| | |Pos| | |Neg| | |Size| | |Pos| | |Neg| | |Size| | Corpus |Size| |
| --- | --- | --- | --- | --- | --- | --- | --- | --- | --- |
| C | NL-Code | 5,196 | 149,000 | 146,852 | 209 | 11,282 | 11,293 | 853 | 787,516 |
| | Code-Code | 4,391 | 122,758 | 145,849 | 193 | 9,162 | 11,282 | 798 | |
| C# | NL-Code | 4,878 | 75,386 | 55,579 | 207 | 6,574 | 4,757 | 828 | 251,147 |
| | Code-Code | 4,397 | 69,016 | 54,886 | 194 | 5,854 | 4,742 | 785 | |
| C++ | NL-Code | 6,181 | 612,647 | 608,088 | 269 | 25,752 | 25,516 | 1,098 | 18,212,508 |
| | Code-Code | 6,181 | 554,465 | 608,088 | 269 | 23,503 | 25,516 | 1,098 | |
| D | NL-Code | 3,359 | 7,624 | 3,655 | 133 | 351 | 142 | 521 | 15,984 |
| | Code-Code | 1,968 | 4,265 | 2,722 | 80 | 218 | 119 | 293 | |
| Go | NL-Code | 3,764 | 25,656 | 18,957 | 165 | 1,466 | 750 | 662 | 68,237 |
| | Code-Code | 3,090 | 21,787 | 18,079 | 148 | 1,242 | 727 | 563 | |
| Haskell | NL-Code | 3,173 | 15,138 | 7,084 | 173 | 2,172 | 936 | 687 | 44,682 |
| | Code-Code | 2,305 | 11,863 | 6,373 | 160 | 1,871 | 922 | 596 | |
| Java | NL-Code | 5,930 | 393,891 | 375,416 | 250 | 17,623 | 16,008 | 1,021 | 2,523,044 |
| | Code-Code | 5,792 | 320,738 | 375,176 | 245 | 14,022 | 15,981 | 991 | |
| Javascript | NL-Code | 2,609 | 15,605 | 13,706 | 134 | 1,322 | 1,345 | 534 | 56,917 |
| | Code-Code | 1,986 | 12,821 | 12,678 | 116 | 1,144 | 1,306 | 436 | |
| Kotlin | NL-Code | 4,017 | 46,487 | 25,600 | 158 | 1,859 | 1,036 | 654 | 121,569 |
| | Code-Code | 3,237 | 39,813 | 24,948 | 127 | 1,600 | 1,009 | 518 | |
| Ocaml | NL-Code | 1,424 | 2,327 | 1,382 | 97 | 219 | 114 | 381 | 7,012 |
| | Code-Code | 485 | 903 | 746 | 50 | 122 | 82 | 170 | |
| PHP | NL-Code | 1,965 | 6,301 | 8,870 | 136 | 896 | 834 | 547 | 29,179 |
| | Code-Code | 1,180 | 4,303 | 6,689 | 99 | 723 | 745 | 389 | |
| Pascal | NL-Code | 4,432 | 113,222 | 105,127 | 216 | 10,113 | 8,568 | 853 | 494,473 |
| | Code-Code | 3,949 | 97,179 | 104,320 | 208 | 8,496 | 8,564 | 816 | |
| Perl | NL-Code | 1,276 | 3,903 | 1,957 | 102 | 559 | 338 | 412 | 11,035 |
| | Code-Code | 678 | 2,627 | 1,531 | 64 | 457 | 305 | 309 | |
| Python | NL-Code | 4,930 | 317,013 | 284,975 | 213 | 17,131 | 15,194 | 859 | 2,290,854 |
| | Code-Code | 4,736 | 266,459 | 284,657 | 210 | 14,144 | 15,192 | 837 | |
| Ruby | NL-Code | 2,349 | 15,230 | 7,278 | 157 | 2,371 | 866 | 649 | 44,934 |
| | Code-Code | 1,742 | 12,714 | 6,683 | 145 | 2,113 | 854 | 569 | |
| Rust | NL-Code | 3,860 | 30,673 | 14,923 | 137 | 742 | 303 | 551 | 59,829 |
| | Code-Code | 3,062 | 26,779 | 14,290 | 104 | 605 | 288 | 428 | |
| Scala | NL-Code | 2,555 | 7,858 | 5,210 | 144 | 867 | 459 | 591 | 24,780 |
| | Code-Code | 1,527 | 5,268 | 4,078 | 123 | 723 | 442 | 448 | |

From each of these problems, we randomly choose one correct solution as a (PL code) query and pair it with the other correct solutions as positive examples and the corresponding wrong solutions (i.e., with execution outcome WRONG ANSWER) as negative examples.

**Retrieval Corpus Metadata and Evaluation Protocol**  We preserve the problem specifications and execution outcomes (e.g., PASSED, WRONG ANSWER) for each candidate code in our retrieval database. For both the NL-code and code-code retrieval tasks, we use this information to determine the correctness of a retrieved code, checking if that solves the same programming problem as the input query by passing all its unit tests or not.

**Evaluation Metrics**  We evaluate the retrieval performance in terms of *retrieval accuracy@k*: computed as the proportion of queries for which a correct code retrieved within top-$k$.

Our retrieval benchmark has 17 programming languages and our training dataset is the largest that provides annotations of similar codes that are found logically equivalent or correct based on the passing of test cases. For evaluation purposes (i.e., for test sets), we release the input problem description (in NL-Code) or the input code (in Code-Code) only and keep all other metadata confidential. Covered programming languages and their data statistics in both tasks are summarized in Table 9.

**Retrieval Evaluation** Figure 12 reports one retrieval task (*code-code*) performance. As anticipated, the *retrieval* capability for the same language pair (a.k.a., monolingual retrieval) of our baseline model performances are relatively stronger and we observe performance degradation when performing cross-lingual retrieval between different languages. However, surprisingly, mono-lingual retrieval accuracies for popular languages like *C*, *C++*, *C#*, *Python*, and *Java* are lower than others such as *Go*, *Haskell*, *Javascript*, *Kotlin*, *Ruby*, *Scala* etc., possibly due to their large retrieval corpus size and presence of more hard negative candidates (very similar to the correct code). Furthermore it is suspected that the lack of enough resource on *D* programming language in both *The Stack* (Kocetkov et al., 2022) and xCODEEVAL is the primary reason for its poor scores.

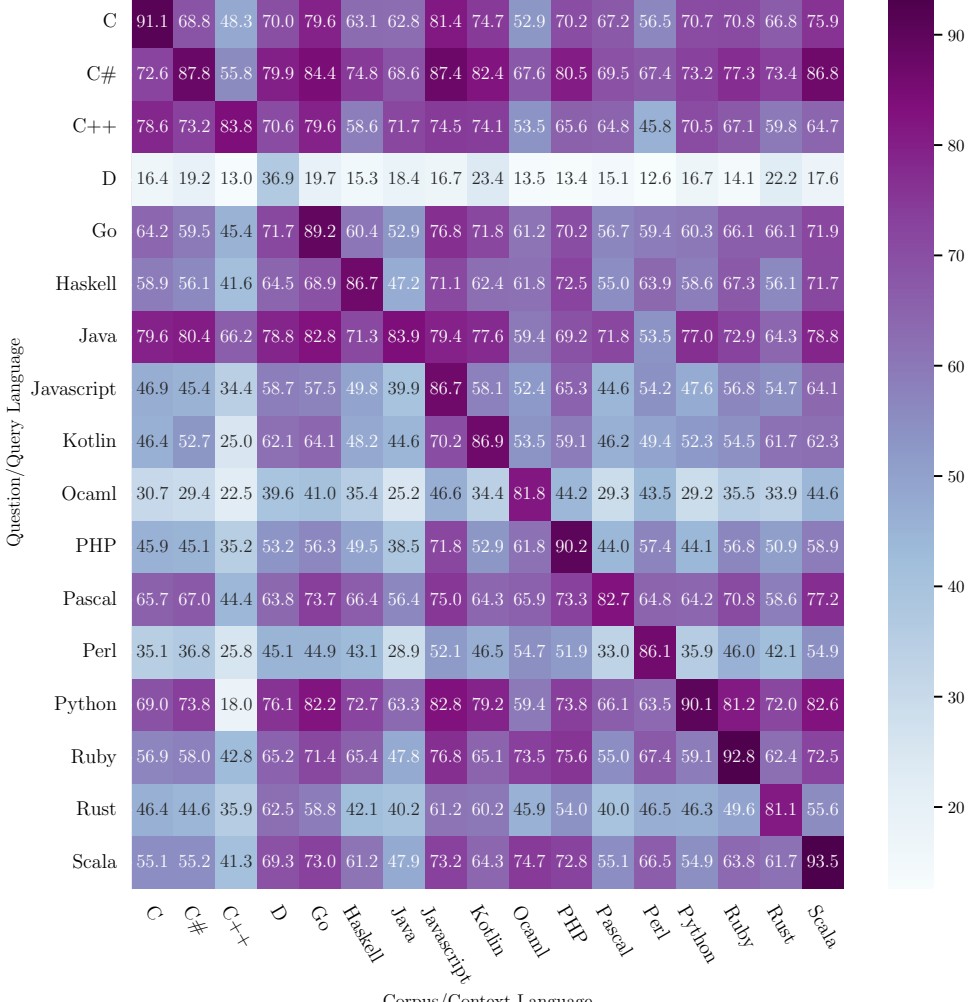

FIGURE 12 – 17 × 17 matrix of top-100 accuracy scores of *StarEncoder* finetuned on retrieval *Code-Code* dataset. Here a cell $(x, y)$ denotes the top-100 accuracy score for code queries from language $x$ and the retrieval corpus of language $y$. The average mono lingual retrieval accuracy is 84.19, and average cross lingual score is 56.93.

```
{
    "C": "GNU C11",
    "C#": "Mono C#",
    "C++": "GNU C++17",
    "Go": "Go",
    "Java": "Java 17",
    "Javascript": "Node.js",
    "Kotlin": "Kotlin 1.4",
    "PHP": "PHP",
    "Python": "PyPy 3",
    "Ruby": "Ruby 3",
    "Rust": "Rust 2018",
}
```

FIGURE 13 – List of `ExecEval` compiler versions used to evaluate the generated codes.

## F  EVALUATION METRICS

**Tag Classification** : Since it is a multi-class multi-label classification problem, we use *f1* score with macro averaging over the classes (in this case the tags) to measure the performance as macro averaging is class population size independent. This is done by first calculating the *f1* score for each class (tag) $T \in \mathcal{T}$ (the set of all tags) with the following formula Taha and Hanbury (2015)

$$\text{f1}_T = \frac{2 * \text{Precision}_T * \text{Recall}_T}{\text{Precision}_T + \text{Recall}_T} = \frac{2 * \text{TP}_T}{2 * \text{TP}_T + \text{FP}_T + \text{FN}_T}$$

And then the macro average is calculated as the mean of $\text{f1}_T$ for all $T \in \mathcal{T}$ Opitz and Burst (2019).

$$\text{f1}_{\text{macro}} = \frac{1}{|\mathcal{T}|} \sum_{T \in \mathcal{T}} \text{f1}_T.$$

**Code Compilation** : Since it is a binary classification problem, we use *accuracy* which is defined as the proportion of correct prediction among all predictions Metz (1978). That is

$$\text{Accuracy} = \frac{\text{TP} + \text{TN}}{\text{TP} + \text{TN} + \text{FP} + \text{FN}}.$$

**Generative Tasks** : The generative tasks in XCODEEVAL(i.e. *Automatic Program Repair*, *Code Translation*, *Program Synthesis*) are all evaluated using pass@$k$ used in Chen et al. (2021).

**Code Retrieval** : The *Code-Code*, and *NL-Code* retrieval tasks in XCODEEVALis evaluated using top-$k$ accuracy (Thakur et al., 2021).

## G  IMPLEMENTATION DETAILS

**Classification tasks** : OpenAI chat completion API with gpt-3.5-turbo-0301 model was used at temperature 0.325 and $n = 1$. List prompts for *Code2Tag*, *DescCode2Tag*, *Code Compilation* as figure or inline styling, with example api response. Then evaluate each through corresponding metric as mentioned in *Appendix F*.

**Generative tasks** : OpenAI chat completion API with gpt-3.5-turbo-0301 model was used at temperature $np.linspace(0, 2, 20)$ and $n = 1$ for *Program Synthesis*. Then upon identifying best temperature at 0.325, another batch of codes were generated at temperature 0.325 and $n = 20$. For *APR*, *Code Translation* temperature of 0.325 and $n = 10$ was used. The generated codes were executed with `ExecEval` with default parameters (follow appendix H for a list of different parameters and their default values) to determine its functional correctness and then evaluated using pass@$k$. Figure 13 shows the compiler versions used to execute the generated codes.

**Retrieval tasks** : We finetuned a DPR [12] model with starencoder for both query and corpus encoder. Both *NL-Code*, and *Code-Code* were trained with maximum sequence length of 1024, and effective batch size 48 for 37 epochs. The model is trained with a multilingual manner. For *Code-Code* we used xCODEEVAL as it is, and for *NL-Code* we made the following template: 'Description: {{description}} Input specification: {{input_spec}} Output specification: {{output_spec}}' for the query string. For evaluation we used corpus provided by xCODEEVAL to generate the dense vectors and then perform queries with test split for both *Code-Code*, and *NL-Code*. Finally the top-$k$ accuracies were measured.

## H EXECEVAL DETAILS

TABLE 10 – Supported languages and their compiler/interpreter versions of our dataset in `ExecEval`.

| Language | Versions |
|---|---|
| Ruby | Ruby 3, Ruby |
| Javascript | Node.js, JavaScript |
| Go | Go 1.19 |
| C++ | GNU C++17, GNU C++17 (64), GNU C++20 (64), GNU C++11, Clang++17 Diagnostics, GNU C++, GNU C++14, GNU C++17 Diagnostics, Clang++20 Diagnostics, MS C++, GNU C++0x, MS C++ 2017 |
| C | GNU C11, GNU C |
| Java | Java 6, Java 7, Java 17, Java 11, Java 8 |
| Python | PyPy 3, PyPy 3-64, Python 3 + libs, Python 2, PyPy 2, Python 3 |
| C# | MS C#, C# 8, Mono C#, .NET Core C#, C# 10 |
| PHP | PHP 8.1 |
| Rust | Rust, Rust 2021 |
| Kotlin | Kotlin, Kotlin 1.4, Kotlin 1.5, Kotlin 1.7, Kotlin 1.6 |

`ExecEval` is an automated code execution and evaluation engine distributed through docker for security and portability. It supports 44 compiler versions for 11 programming languages as shown in table 10. It exposed `NUM_WORKERS` CLI argument to spawn multiple workers that can execute the codes. It is highly scalable in the sense of adding support for more languages or one can just change `NUM_WORKERS` to execute more codes in parallel. At the top level of `ExecEval`, there is a HTTP server that exposes 2 API endpoints */api/execute_code*, */api/all_runtimes*. Figure 15 shows a simple usage of *execute_code* API. By default the execution of a code is stopped when the code doesn't pass a unit test as pass@$k$ depends on whether *all* the unit tests passed or not. This can be disabled by adding '"stop_at_first_fail": false', in which case all unit tests for a given code will be executed irrespective of the outcomes for other unit tests. Figure 6 is generated with disabling 'stop_at_first_fail'. It is worth noting that, disabling this setting can increase the evaluation time significantly (e.g. in table 3 for *program synthesis (N)* where *23,320* codes were executed the difference was of approximately *12 minutes* and *2 hours 12 minutes* where `ExecEval` was running with 61 workers).

**Security Measures** `ExecEval` uses `prlimit` [13], and `seccomp` [14] to limit system resources allocated for any instance of code executed through the API endpoint in addition to using unique unprivileged users for each worker spawned with *NUM_WORKERS*. Table 11 shows the default values provided to `prlimit`, furthermore *nofile*, and *nproc* are customized for each of the supported languages. The *seccomp* is used to block *socket* system call, which disables network access (this is default). One can enable network access by adding '"block_network": false' in the request body as shown in fig. 15. Similarly, adding a 'limits' object in the request body allows one to change the

---

12. https://github.com/facebookresearch/DPR
13. https://man7.org/linux/man-pages/man1/prlimit.1.html
14. https://man7.org/linux/man-pages/man2/seccomp.2.html

```json
{
    "compile_cmd": "node",
    "compile_flags": "--check",
    "execute_cmd": "node",
    "execute_flags": "",
    "has_sanitizer": false,
    "is_compiled": true,
    "runtime_name": "JavaScript",
    "timelimit_factor": 3
}
```

FIGURE 14 – An example runtime object the response from `/api/all_runtimes` contains list of such objects.

```json
{
  "language": "Python 3",
  "source_code": "a, b = map(↩
      ↪ int, input().strip().↩
      ↪ split())\nprint(a+b)"↩
      ↪ ,
  "unittests": [
    {"input": "1 1", "output":↩
        ↪ ["2"]},
    {"input": "1 10", "output"↩
        ↪ : ["11"]}
  ]
}
```

```json
{
  "data": [
    {
      "exec_outcome": "PASSED"↩
          ↪ ,
      "input": "1 1",
      "output": [
        "2"
      ],
      "result": "2"
    },
    {
      "exec_outcome": "PASSED"↩
          ↪ ,
      "input": "1 10",
      "output": [
        "11"
      ],
      "result": "11"
    }
  ]
}
```

FIGURE 15 – On left: An example request body for `/api/execute_code` The Python code takes 2 numbers as input and prints their sum. On right: The response by `ExecEval` in response to the request shown in left.

TABLE 11 – Default resource limits values for `prlimit` used by `ExecEval`. The comment column shows the variable names as defined in `sys/resource.h` with some additional information.

| Resource | Value | Comment |
|---|---|---|
| core | 0 | RLIMIT_CORE |
| data | -1 | RLIMIT_DATA |
| fsize | 0 | RLIMIT_FSIZE |
| sigpending | 0 | RLIMIT_SIGPENDING |
| rss | -1 | RLIMIT_RSS |
| nofile | 4 | RLIMIT_NOFILE |
| msgqueue | 0 | RLIMIT_MSGQUEUE |
| rtprio | 0 | RLIMIT_RTPRIO |
| stack | -1 | RLIMIT_STACK |
| cpu | 2 | RLIMIT_CPU, CPU time, in seconds |
| nproc | 1 | RLIMIT_NPROC |
| as | $2 \times 1024^3$ | RLIMIT_AS set to 2GB by default |
| locks | 0 | RLIMIT_LOCKS |

limits for executing an individual code. [15] The execution of code via an unprivileged user disables the read, write, or execute permissions of any sensitive files. Figure 16, 17, and 18 shows an example of a fork bomb written in C, a network request in Python, and an escalated access in Python which are all blocked by `ExecEval`, respectively.

```c
#include <stdio.h>
#include <sys/types.h>

int main()
{
  while(1)
    fork();
  return 0;
}
```

```
{
  "data": [
    {
      "exec_outcome": "↩
        ↪ TIME_LIMIT_EXCEEDED↩
        ↪ ",
      "input": "",
      "output": [
        ""
      ],
      "result": null
    }
  ]
}
```

FIGURE 16 – Left: An fork bomb written in C. Right: `ExecEval` ran the code with allowing only 1 process and thus the infinite loop resulted in `TIME_LIMIT_EXCEEDED`.

---

15. For more details follow: `https://github.com/ntunlp/ExecEval`.

## I    DEFINITION OF DATA ATTRIBUTES

All the raw data can be downloaded from the huggingface [16]. For each of the tasks we have two data files that are required for multiple tasks.

1. `problem_descriptions.jsonl`
2. `unittest_db.json`

You can find these two files in the root directory of the main branch of `huggingface` dataset repository. To avoid data redundancy we didn't include these data with the relevant tasks, rather we add a unique id `src_uid` to retrieve these data. We include a data loader using `datasets` [17] package that defines the tasks.

We provide the definition for each of the data attributes of XCODEEVAL in the following sections.

### I.1    PROBLEM DESCRIPTION (`PROBLEM_DESCRIPTION`)

The problem descriptions are in the `problem_descriptions.jsonl` file. This data source is linked to the proposed tasks by matching the `src_uid` column for each sample in the relevant tasks. The columns copied from the `problem_descriptions.jsonl` file are prefixed with `prob_desc_`.

1. `description`: Problem description in textual format, math operations are written in latex.
2. `input_from`: How the program should take the unit test.
3. `output_to`: Where the program should output the result of the unit test.
4. `time_limit`: Time limit to solve the problem.
5. `memory_limit`: Memory limit to solve the problem.
6. `input_spec`: How and in what order the input will be given to the program? It also includes the date range, types, and sizes.
7. `output_spec`: How the outputs should be printed. Most of the time the unit test results are matched with an *exact string match* or *floating point comparison* with a precision boundary.
8. `sample_inputs`: A sample input for the code that is expected to solve the problem described in `description`.
9. `sample_outputs`: The expected output for the `sample_input` that is expected to solve the problem described in `description`.
10. `notes`: Explanation of `sample_inputs` & `sample_outputs`.
11. `tags`: The problem categories.
12. `src_uid`: The unique id of the problem. This ID is referred to in the task data samples instead of putting all this information.
13. `difficulty`: How difficult is it to solve the problem for a human (annotated by an expert human).
14. `created_at`: The Unix timestamp when the problem was released. Use `datetime` lib in Python to parse it to a human-readable format.

### I.2    UNIT TESTS (`HIDDEN_UNIT_TEST`)

The unit tests needed for execution based evaluation are in the `unittest_db.json` file. This data source is linked to the proposed tasks by matching the `src_uid` column for each sample in the relevant tasks. The columns copied from the `unittest_db.json` file are under the attribute `hidden_unit_test`.

1. `unittest_db.json` dict keys i.e., `db884d679d9cfb1dc4bc511f83beedda` are the `src_uid` from `problem_descriptions.jsonl`.
2. `input`: Input of the unit test.
3. `output`: List of expected outputs for the unit test.

---

16. https://huggingface.co/datasets/NTU-NLP-sg/xCodeEval
17. https://github.com/huggingface/datasets

### I.3 TAG CLASSIFICATION (`TAG_CLASSIFICATION`)

Given a `source_code` the objective is to classify the code into multi-label tags (label:`tags`).

1. `lang`: Runtime/compiler version of the `source_code`.
2. `source_code`: A program.
3. `tags`: List of potential algorithmic techniques required to write the program.
4. `lang_cluster`: A generic programming language name the value of `lang` belongs to.
5. `code_uid`: A unique ID for the sample. It is not important for model training. If you find any issue with the sample, you can report it to us by mentioning the `code_uid`.
6. `src_uid`: A specific identifier that shows which problem the code is associated with. This identifier is **important** for the training of the model. The problem referred to by the `src_uid` provides a natural description of the problem that the code successfully solved. Refer to Structure of `problem_descriptions.jsonl`.
7. `difficulty`: Difficulty rating of the problem indicated by `src_uid`. The higher the harder.

### I.4 CODE COMPILATION (`CODE_COMPILATION`)

Given a `source_code` the objective is to classify if the code compiles or not (label:`compilation_error`).

1. `lang`: Runtime/Compiler version of the `source_code`.
2. `source_code`: A program.
3. `lang_cluster`: A generic programming language name the value of `lang` belongs to.
4. `compilation_error`: True/False, Indicates if the code generates a compilation error or not.
5. `code_uid`: A unique ID for the sample. It is not important for model training. If you find any issue with the sample, you can report it to us by mentioning the `code_uid`.
6. `src_uid`: A specific identifier that shows which problem the code is associated with. This identifier is **important** for the training of the model. The problem referred to by the `src_uid` provides a natural description of the problem that the code successfully solved. Refer to Structure of `problem_descriptions.jsonl`.
7. `difficulty`: Difficulty rating of the problem indicated by `src_uid`. The higher the harder.
8. `file_name`: Name of the source JSON file from where data is loaded.

### I.5 AUTOMATIC PROGRAM REPAIR (`APR`)

Given a `bug_source_code` the objective is to generate a fixed version of the code that passes all the unit tests. Use `fix_source_code` for training.

1. `similarity_score`: A similarity score between `bug_source_code` and `fix_source_code` given by difflib.
2. `equal_cnt`: A metric comparing `bug_source_code` and `fix_source_code`. Recommended by difflib.
3. `replace_cnt`: A metric comparing `bug_source_code` and `fix_source_code`. Recommended by difflib.
4. `delete_cnt`: A metric comparing `bug_source_code` and `fix_source_code`. Recommended by difflib.
5. `insert_cnt`: A metric comparing `bug_source_code` and `fix_source_code`. Recommended by difflib.
6. `fix_ops_cnt`: A metric comparing `bug_source_code` and `fix_source_code`. Recommended by difflib.
7. `bug_source_code`: Buggy code.
8. `fix_source_code`: A potential fix of the buggy code that passed all the unit tests.

9. `lang`: Runtime/Compiler version of the `source_code`.

10. `fix_code_uid`: A unique ID for the fix code. It is not important for model training. If you find any issue with the sample, you can report it to us mentioning the `fix_code_uid`.

11. `bug_code_uid`: A unique ID for the buggy code. It is not important for model training. If you find any issue with the sample, you can report it to us mentioning the `bug_code_uid`.

12. `src_uid`: A specific identifier that shows which problem the code is associated with. This identifier is **important** for the training of the model. The problem referred to by the `src_uid` provides a natural description of the problem that the code successfully solved. Refer to Structure of `problem_descriptions.jsonl`.

13. `apr_id`: A unique ID for the apr sample. It is not important for model training. If you find any issue with the sample, you can report it to us mentioning the `apr_id`.

14. `difficulty`: Difficulty rating of the problem indicated by `src_uid`. The higher the harder.

15. `tags`: List of potential algorithmic techniques required to write the program.

16. `bug_exec_outcome`: A pre-run execution outcome of `bug_source_code`. Follow Section 2.2 to know the potential list of outcomes. The `exec_outcome` flags in the training data comes from a pre-run environment from the source website and they are not verified in **ExecEval**. However, training data doesn't include unit-test to avoid potential hacks and confusion. We provide unit test for only validation and test data.

17. `fix_exec_outcome`: A pre-run execution outcome of `fix_source_code`. Follow Section 2.2 to know the potential list of outcomes. The `exec_outcome` flags in the training data comes from a pre-run environmentfrom the source website and they are not verified in **ExecEval**. However, training data doesn't include unit-test to avoid potential hacks and confusion. We provide unit test for only validation and test data.

18. `potential_dominant_fix_op`: A potential fix op recommended by difflib.

19. `lang_cluster`: A generic programming language name the value of `lang` belongs to.

20. `prob_desc_description`: Problem description in textual format, math operations are written in latex.

21. `prob_desc_input_from`: How the program should take the unit test.

22. `prob_desc_output_to`: Where the program should output the result of the unit test.

23. `prob_desc_time_limit`: Time limit to solve the problem.

24. `prob_desc_memory_limit`: Memory limit to solve the problem.

25. `prob_desc_input_spec`: How and in what order the input will be given to the program? It also includes the date range, types, and sizes.

26. `prob_desc_output_spec`: How the outputs should be printed. Most of the time the unit test results are matched with an *exact string match* or *floating point comparison* with a precision boundary.

27. `prob_desc_sample_inputs`: A sample input for the code that is expected to solve the problem described in `description`.

28. `prob_desc_sample_outputs`: The expected output for the `sample_input` that is expected to solve the problem described in `description`.

29. `prob_desc_notes`: Explanation of `sample_inputs` & `sample_outputs`.

30. `prob_desc_created_at`: The Unix timestamp when the problem was released. Use `datetime` lib in Python to parse it to a human-readable format.

31. `file_name`: Name of the source jsonl file from where data is loaded.

32. `hidden_unit_tests`: a list of unit tests returned as string. use `json.loads(hidden_unit_tests)` to load the data.

## I.6 CODE TRANSLATION (`CODE_TRANSLATION`)

Given a source code (`source_code`) in `lang_cluster`, generate a code in target programming language.

1. `lang`: Runtime/Compiler version of the `source_code`.

2. `source_code`: A program.

3. `code_uid`: A unique ID for the sample. It is not important for model training. If you find any issue with the sample, you can report it to us by mentioning the `code_uid`.

4. `src_uid`: A specific identifier that shows which problem the code is associated with. This identifier is **important** for the training of the model. The problem referred to by the `src_uid` provides a natural description of the problem that the code successfully solved. Refer to Structure of `problem_descriptions.jsonl`

5. `difficulty`: Difficulty rating of the problem indicated by `src_uid`. The higher the harder.

6. `exec_outcome`: Execution outcome status. Follow Section 2.2 to know the potential list of outcomes. The `exec_outcome` flags in the training data comes from a pre-run environment from the source website and they are not verified in **ExecEval**. However, training data doesn't include unit-test to avoid potential hacks and confusion. We provide unit test for only validation and test data

7. `lang_cluster`: A generic programming language name the value of `lang` belongs to.

8. `prob_desc_description`: Problem description in textual format, math operations are written in latex.

9. `prob_desc_input_from`: How the program should take the unit test.

10. `prob_desc_output_to`: Where the program should output the result of the unit test.

11. `prob_desc_time_limit`: Time limit to solve the problem.

12. `prob_desc_memory_limit`: Memory limit to solve the problem.

13. `prob_desc_input_spec`: How and in what order the input will be given to the program? It also includes the date range, types, and sizes.

14. `prob_desc_output_spec`: How the outputs should be printed. Most of the time the unit test results are matched with an *exact string match* or *floating point comparison* with a precision boundary.

15. `prob_desc_sample_inputs`: A sample input for the code that is expected to solve the problem described in `description`.

16. `prob_desc_sample_outputs`: The expected output for the `sample_input` that is expected to solve the problem described in `description`.

17. `prob_desc_notes`: Explanation of `sample_inputs` & `sample_outputs`.

18. `prob_desc_created_at`: The Unix timestamp when the problem was released. Use `datetime` lib in Python to parse it to a human-readable format.

19. `file_name`: Name of the source jsonl file from where data is loaded.

20. `hidden_unit_tests`: a list of unit tests returned as string. use `json.loads(hidden_unit_tests)` to load the data.

## I.7 PROGRAM SYNTHESIS (`PROGRAM_SYNTHESIS`)

Given a `src_uid` read problem description from `problem_descriptions.jsonl` and generate a solution for problem description.

1. `lang`: Runtime/Compiler version of the `source_code`.

2. `source_code`: A program.

3. `code_uid`: A unique ID for the sample. It is not important for model training. If you find any issue with the sample, you can report it to us by mentioning the `code_uid`.

4. `src_uid`: A specific identifier that shows which problem the code is associated with. This identifier is **important** for the training of the model. The problem referred to by the `src_uid` provides a natural description of the problem that the code successfully solved. Refer to Structure of `problem_descriptions.jsonl`.

5. `difficulty`: Difficulty rating of the problem indicated by `src_uid`. The higher the harder.

6. `exec_outcome`: Execution outcome status. Follow Section 2.2 to know the potential list of outcomes. The `exec_outcome` flags in the training data comes from a pre-run environment. However, training data doesn't include unit-test to avoid potential hacks. We provide unit tests for only dev and test data.

7. `lang_cluster`: A generic programming language name the value of `lang` belongs to.
8. `prob_desc_description`: Problem description in textual format, math operations are written in latex.
9. `prob_desc_input_from`: How the program should take the unit test.
10. `prob_desc_output_to`: Where the program should output the result of the unit test.
11. `prob_desc_time_limit`: Time limit to solve the problem.
12. `prob_desc_memory_limit`: Memory limit to solve the problem.
13. `prob_desc_input_spec`: How and in what order the input will be given to the program? It also includes the date range, types, and sizes.
14. `prob_desc_output_spec`: How the outputs should be printed. Most of the time the unit test results are matched with an *exact string match* or *floating point comparison* with a precision boundary.
15. `prob_desc_sample_inputs`: A sample input for the code that is expected to solve the problem described in `description`.
16. `prob_desc_sample_outputs`: The expected output for the `sample_input` that is expected to solve the problem described in `description`.
17. `prob_desc_notes`: Explanation of `sample_inputs` & `sample_outputs`.
18. `prob_desc_created_at`: The Unix timestamp when the problem was released. Use `datetime` lib in Python to parse it to a human-readable format.
19. `file_name`: Name of the source jsonl file from where data is loaded.
20. `hidden_unit_tests`: a list of unit tests returned as a string. use `json.loads(hidden_unit_tests)` to load the data.

## I.8 RETRIEVAL CORPUS (`RETRIEVAL_CORPUS`)

Use the `retrieval_corpus` to perform query for `retrieval_nl_code` (appendix I.9) and `retrieval_code_code` (appendix I.10).

1. `idx`: An integer index to identify the code. It is unique within the codes of each language.
2. `source_code`: A program.
3. `file_name`: Name of the source jsonl file from where data is loaded.

## I.9 RETRIEVAL NL-CODE (`RETRIEVAL_NL_CODE`)

Given a NL (problem description) retrieve similar source code from `retrieval_corpus` (appendix I.8).

1. `nl` : Problem description in textual format, math operations are written in latex. Given as input query.
2. `positive_code` : list of positive codes for `nl`.
3. `negative_code` : list of negative codes for `nl`.
4. `src_uid` : A specific identifier that shows which problem the code is associated with. This identifier is **important** for the training of the model. The problem referred to by the `src_uid` provides a natural description of the problem that the code successfully solved. Refer to Structure of `problem_descriptions.jsonl`.
5. `file_name`: Name of the source jsonl file from where data is loaded.

## I.10 RETRIEVAL CODE-CODE (`RETRIEVAL_CODE_CODE`)

Given a `source_code`, retrieve similar source code from `retrieval_corpus` (appendix I.8).

1. `positive_code` : list of positive codes for `nl`.
2. `negative_code` : list of negative codes for `nl`.
3. `src_uid` : A specific identifier that shows which problem the code is associated with. This identifier is **important** for the training of the model. The problem referred to by the `src_uid` provides a natural description of the problem that the code successfully solved. Refer to Structure of `problem_descriptions.jsonl`.

4. `source_code`: A source code given as input query.
5. `file_name`: Name of the source jsonl file from where data is loaded.

## J DATASHEETS FOR DATASETS

We follow the questionnaires from Gebru et al. (2021) as the datasheet for XCODEEVAL .

### J.1 MOTIVATION

**For what purpose was the dataset created?**  XCODEEVAL dataset was specifically created to address three main aspects: (i) *Reasoning*, (ii) *Multilinguality* in terms of programming languages, and (iii) *Executability* of the programming languages. These aspects were thoroughly discussed in Section 1 of the main paper, providing detailed insights into the motivation behind the dataset creation.

**Who created the dataset (e.g., which team, research group) and on behalf of which entity (e.g., company, institution, organization)?**  XCODEEVAL is an output of a passion project driven by a group of researchers from (i) *Islamic University of Technology* (ii) *Nanyang Technological University* (iii) *Bosch Research*.

**Who funded the creation of the dataset?**  *Nanyang Technological University* provided the necessary computing resources for the project. None of the project members received any remuneration for their contributions.

### J.2 COMPOSITION

**What do the instances that comprise the dataset represent (e.g., documents, photos, people, countries)?**  Please follow the Section I for details.

**How many instances are there in total (of each type, if appropriate)?**  Please follow the Table 8, 2, and 9 for the details statistics of the dataset.

**Does the dataset contain all possible instances or is it a sample (not necessarily random) of instances from a larger set?**  Dataset contains all possible instances.

**What data does each instance consist of?**  Please follow the Section I for details.

**Is there a label or target associated with each instance?**  Please follow the Section I for details.

**Is any information missing from individual instances?**  For a few problem description, `difficulty` is assigned as `None` due to data unavailability.

**Are relationships between individual instances made explicit (e.g., users' movie ratings, social network links)?**  Please follow the Section I for details.

**Are there recommended data splits (e.g., training, development/validation, testing)?**  We explicitly defined the training, validation and test split for XCODEEVAL . Please follow the Section 2.1 for more details.

**Are there any errors, sources of noise, or redundancies in the dataset?**  To the best of our knowledge there are not errors, sources of noise or redundencies in XCODEEVAL .

**Is the dataset self-contained, or does it link to or otherwise rely on external resources (e.g., websites, tweets, other datasets)**  The dataset is self-contained.

**Does the dataset contain data that might be considered confidential (e.g., data that is protected by legal privilege or by doctor– patient confidentiality, data that includes the content of individuals' non-public communications)?** The dataset is collected from open sourced sources. There are no confidentiality or non-public entity in the dataset.

**Does the dataset contain data that, if viewed directly, might be offensive, insulting, threatening, or might otherwise cause anxiety?** To the best of our knowledge there are no *offensive*, *insulting*, *threatening* content in the dataset.

**Does the dataset identify any subpopulations (e.g., by age, gender)?** No.

**Is it possible to identify individuals (i.e., one or more natural persons), either directly or indirectly (i.e., in combination with other data) from the dataset?** There are no attributes in the dataset that allow to identify individuals.

**Does the dataset contain data that might be considered sensitive in any way (e.g., data that reveals race or ethnic origins, sexual orientations, religious beliefs, political opinions or union memberships, or locations; financial or health data; biometric or genetic data; forms of government identification, such as social security numbers; criminal history)?** There are no attributes in the dataset that allow this.

### J.3    COLLECTION PROCESS

**How was the data associated with each instance acquired?** Following Li et al. (2022), the data was collected from `codeforces.com` and then associated with different tasks. Please follow Section E for more details.

**If the dataset is a sample from a larger set, what was the sampling strategy (e.g., deterministic, probabilistic with specific sampling probabilities)?** Dataset wasn't sampled from a large dataset. We proposed the dataset for the first time.

**Who was involved in the data collection process (e.g., students, crowdworkers, contractors) and how were they compensated (e.g., how much were crowdworkers paid)?** Dataset was collected by the author of this paper.

**Over what timeframe was the data collected?** The data was downloaded in between `Feb, 2022` to `January, 2023`.

**Were any ethical review processes conducted (e.g., by an institutional review board)?** No.

**Did you collect the data from the individuals in question directly, or obtain it via third parties or other sources (e.g., websites)?** The data is downloaded by an author. No third parties or other sources are involved.

**Has an analysis of the potential impact of the dataset and its use on data subjects (e.g., a data protection impact analysis) been conducted?** Potential impact of the dataset is discussed at section 5 and section 6.

### J.4    PREPROCESSING/CLEANING/LABELING

**Was any preprocessing/cleaning/labeling of the data done (e.g., discretization or bucketing, tokenization, part-of-speech tagging, SIFT feature extraction, removal of instances, processing of missing values)?** We de-anonymized the data and remove data with sensitive information (i.e., email, large infograph, toxic keywords). The labels come as a metadata from the sources.

**Was the "raw" data saved in addition to the preprocessed/cleaned/labeled data (e.g., to support unanticipated future uses)?** No.

**Is the software that was used to preprocess/clean/label the data available?** No software was used for labeling the data.

**Has the dataset been used for any tasks already?** Yes. We evaluated ChatGPT and trained *StarEncoder* using the dataset.

**Is there a repository that links to any or all papers or systems that use the dataset** Yes, `https://github.com/ntunlp/xCodeEval`.

**What (other) tasks could the dataset be used for?** We proposed 7 different tasks for xCodeE-val . Please follow table 8 for details.

**Is there anything about the composition of the dataset or the way it was collected and preprocessed/cleaned/labeled that might impact future uses?** No.

## J.5 Distribution

**Will the dataset be distributed to third parties outside of the entity (e.g., company, institution, organization) on behalf of which the dataset was created?** Please follow the Licensing section in `https://github.com/ntunlp/xCodeEval` for details.

**When will the dataset be distributed?** The data is already distributed via huggingface [18].

**Will the dataset be distributed under a copyright or other intellectual property (IP) license, and/or under applicable terms of use (ToU)? I** Please follow the Licensing section in `https://github.com/ntunlp/xCodeEval` for details.

**Have any third parties imposed IP-based or other restrictions on the data associated with the instances?** No.

**Do any export controls or other regulatory restrictions apply to the dataset or to individual instances?** No.

## J.6 Maintenance

**Who will be supporting/hosting/maintaining the dataset?** Huggingface is hosting the dataset. The authors are maintaining the dataset. *Nanyang Technological University* is supporting the dataset.

**How can the owner/curator/manager of the dataset be contacted (e.g., email address)?** Email.

**Is there an erratum?** None at this point. The dataset is hosted through git lfs. The future erratum can be tracked easily.

**Will the dataset be updated (e.g., to correct labeling errors, add new instances, delete instances)?** To the best of our knowledge there are no errors in the dataset. The authors do not intend to add new instances at this point. But the authors remain open to remove/correct instances given that any labeling errors found.

**If the dataset relates to people, are there applicable limits on the retention of the data associated with the instances (e.g., were the individuals in question told that their data would be retained for a fixed period of time and then deleted)?** The dataset doesn't relate to people.

**Will older versions of the dataset continue to be supported/hosted/maintained?** Yes. The older version should be accessed via git LFS.

---

18. https://huggingface.co/datasets/NTU-NLP-sg/xCodeEval

**If others want to extend/augment/build on/contribute to the dataset, is there a mechanism for them to do so?** Since the dataset is fixed, there is currently no way to contribute to it. Please note that any extensions or augmentations of the dataset are subject to the same license as this dataset.

## K    DISCUSSION ON THE POSSIBILITY OF DATA LEAKAGE AND CONTAMINATION

Although we completely separate our validation and test data, LLMs might have possible data leakage from pretraining. We find that even identifying data leakage (test data exists or not in the prertraining corpora) is challenging using conventional data search methods due to search cost & complexity (e.g., exact match or token overlapping methods) while hashing based searches suffer from not having properly segmented text. For leakage-free evaluation, we approach employs "knowledge cut-off" which show that the data contamination significantly impacts the model performance and it needs to be interpreted with proper analyses. We plan to evaluate on seperate human written testset in future.

## L    THE DATASET NUTRITION LABEL

We follow the framework proposed by Holland et al. (2018). Table 12 gives an overview of dataset facts. The variable description can be found in appendix I. We discuss about provenance in appendix J.3 and appendix J.6.

## M    DATA CARD

*Data card* for the XCODEEVAL is distributed via huggingface platform. [Link]

TABLE 12 – Dataset facts for XCODEEVAL . It covers the metadata related to the whole dataset.

| Metadata | |
|---|---|
| Filename | File names for each of the tasks can be found here . |
| Format | jsonl, json, arrow dataset loader. |
| Url | `https://huggingface.co/datasets/NTU-NLP-sg/xCodeEval` |
| Domain | Programming Language, Competitive Programming |
| Keywords | programming-language, code, program-synthesis, automatic-code-repair, code-retrieval, code-translation, code-classification, execution, benchmark, multilingual, multitask, unit-test |
| Type | columnar |
| Rows | Follow table 2, table 8, and table 9 |
| Columns | Follow appendix I |
| Missing | 0% |
| License | CC BY-NC 4.0 |
| Released | MARCH 2023 |
| Range | From `Feb 19, 2010` to `Nov 21, 2022` |
| Description | We introduce xCodeEval, the largest executable multilingual multitask benchmark to date consisting of 25 M document-level coding examples from about 7.5 K unique problems covering up to 17 programming languages with execution-level parallelism. It features a total of seven tasks involving code understanding, generation, translation and retrieval, and it employs an execution-based evaluation. We develop a test-case based multilingual code execution engine, ExecEval that supports all the programming languages in xCodeEval. We also propose a novel data splitting and a data selection schema for balancing data distributions over multiple attributes based on geometric mean and graph-theoretic principle. |

```python
import urllib.request

url = 'http://icanhazip.com'

with urllib.request.urlopen(↩
    ↪ url) as response:
    if response.getcode() == ↩
        ↪ 200:
        print(response.read().↩
            ↪ decode('utf-8')↩
            ↪ .strip())
    else:
        print(f'Request␣failed↩
            ↪ ␣with␣status␣↩
            ↪ code:␣{response↩
            ↪ .getcode()}')
```

```
{
  "data": [
    {
      "exec_outcome": "↩
        ↪ RUNTIME_ERROR",
      "input": "",
      "output": [
        ""
      ],
      "result": "Traceback (↩
        ↪ most recent call ↩
        ↪ last):\n  File \"↩
        ↪ /usr/lib/python3.↩
        ↪ 11/urllib/request↩
        ↪ .py\", line 1348,↩
        ↪  in do_open\n   ↩
        ↪ h.request(req.↩
        ↪ get_method(), req↩
        ↪ .selector, req.↩
        ↪ data, headers,\n ↩
        ↪  File \"/usr/lib/↩
        ↪ python3.11/http/↩
        ↪ client.py\", line↩
        ↪  1282, in request↩
        ↪ \n    self.↩
        ↪ _send_request(↩
        ↪ method, url, body↩
        ↪ , headers, ↩
        ↪ encode_chunked)\n↩
        ↪  **Truncated**
      line 941, in connect\n ↩
        ↪    self.sock = ↩
        ↪ self.↩
        ↪ _create_connection↩
        ↪ (\n ↩
        ↪                 ↩
        ↪ ^^^^^^^^^^^^^^^^^^^^^^^^^\↩
        ↪ n  File \"/usr/↩
        ↪ lib/python3.11/↩
        ↪ socket.py\", line↩
        ↪  826, in ↩
        ↪ create_connection↩
        ↪ \n    for res in ↩
        ↪ getaddrinfo(host,↩
        ↪  port, 0, ↩
        ↪ SOCK_STREAM):\↩
        ↪ nFile \"/usr/lib/↩
        ↪ python3.11/socket↩
        ↪ .py\", line 961, ↩
        ↪ in getaddrinfo\n ↩
        ↪     for res in ↩
        ↪ _socket.↩
        ↪ getaddrinfo(host,↩
        ↪  port, family, ↩
        ↪ type, proto, ↩
        ↪ flags):\nsocket.↩
        ↪ gaierror: [Errno ↩
        ↪ -3] Temporary ↩
        ↪ failure in name ↩
        ↪ resolution"
    }
  ]
}
```

FIGURE 17 – Left: A python code performing a network request. Right: `ExecEval` responded with RUNTIME_ERROR as the *socket* system call is blocked.

```
import subprocess

# Run 'ps -ef' command
command = ['ps', '-ef']
process = subprocess.Popen(↩
    ↪ command, stdout=↩
    ↪ subprocess.PIPE, stderr↩
    ↪ =subprocess.PIPE)
output, error = process.↩
    ↪ communicate()

# Decode and print the output
if output:
    print(output.decode('utf-8↩
        ↪ '))
else:
    print(f'Error:␣{error.↩
        ↪ decode("utf-8")}')
```

```
{
  "data": [
    {
      "exec_outcome": "↩
          ↪ RUNTIME_ERROR",
      "input": "",
      "output": [
        ""
      ],
      "result": "Traceback (↩
          ↪ most recent call ↩
          ↪ last):\n  File \"↩
          ↪ /code_store/6cd9b↩
          ↪ 5215a524abab3712↩
          ↪ bc897de2be5/test.↩
          ↪ py\", line 5, in ↩
          ↪ <module>\n    ↩
          ↪ process = ↩
          ↪ subprocess.Popen(↩
          ↪ command, stdout=↩
          ↪ subprocess.PIPE, ↩
          ↪ stderr=subprocess↩
          ↪ .PIPE)\n ↩
          ↪           ↩
          ↪ ^^^^^^^^^^\n  ↩
          ↪ File \"/usr/lib/↩
          ↪ python3.11/↩
          ↪ subprocess.py\", ↩
          ↪ line 890, in ↩
          ↪ __init__\n    ↩
          ↪ errread, errwrite↩
          ↪ ) = self.↩
          ↪ _get_handles(↩
          ↪ stdin, stdout, ↩
          ↪ stderr)\n ↩
          ↪                 ↩
          ↪ ^^^^^^^^^^^^^^^\n↩
          ↪   File \"/usr/lib↩
          ↪ /python3.11/↩
          ↪ subprocess.py\", ↩
          ↪ line 1664, in ↩
          ↪ _get_handles\n ↩
          ↪     c2pread, ↩
          ↪ c2pwrite = os.↩
          ↪ pipe()\n ↩
          ↪                 ↩
          ↪ ^^^^^^^^^\↩
          ↪ nOSError: [Errno ↩
          ↪ 24] Too many open↩
          ↪  files\n"
    }
  ]
}
```

FIGURE 18 – Left: A python code performing a subprocess call to run 'ps -ef'. Right: `ExecEval` responded with RUNTIME_ERROR as *nofile* (table 11) is limiting the execution of such codes.

