# OpenReview forum: "xCodeEval: An Execution based Large Scale Multilingual Multitask Benchmark for Code Understanding, Generation, Translation and Retrieval"
_ICLR.cc/2024/Conference — Submitted to ICLR 2024_

### Official Review · Reviewer_eCtw · 2023-10-29

**Soundness:** 3 good
**Presentation:** 2 fair
**Contribution:** 3 good
**Rating:** 6
**Confidence:** 4

**Summary:**

This paper introduces a multi-programming-language code dataset based on algorithmic problems and natural language descriptions taken from codeforces.com. The dataset includes 17 languages, with 7.5K problems, where problems have solutions in multiple languages. The problems also have test cases, 63K in total, which can be used in program synthesis (code generation) tasks to verify the correctness of model-generated solutions. The dataset also supports code translation, classification (tagging, executability), retrieval (text<->code or multilingual code<->code retrieval), and program repair. The paper introduces a Docker-based execution testing environment, ExecEval, for the languages and problems. Finally, the paper evaluates ChatGPT, a fine-tuned StarCoder-3B model, and Code-Llama variants on the various tasks from the dataset, finding that performance varies across languages and ChatGPT performance is substantially higher than the other models.

**Strengths:**

S1) The paper uses execution-based testing for its program synthesis tasks, which is important when generating long/complex code, with multiple correct solutions, like the algorithmic tasks here.

S2) The paper's dataset is large-scale and consists of >10 programming languages. For the program synthesis task, this fills a gap in existing execution-based datasets that are either smaller-scale (e.g. MBXP, MultiPL-E) or consist of only a few languages (e.g. APPS, CodeContests).

S3) The EvalExec framework for executing generated code will likely be useful for other work constructing execution-based benchmarks. I appreciated the information in the Appendix about the API server, instrumentation of failure cases, and security measures / configurable resource limits.

S4) While the paper's contribution was mostly on the dataset side, some of the experimental analysis was also interesting, in particular the effect of ChatGPT's knowledge cutoff on performance.

**Weaknesses:**

W1) The contribution of the paper is spread a bit thin, in my opinion. On the dataset side, I think that the dataset will likely be useful for program synthesis, but given the existence of other similar datasets (although they either have fewer languages or smaller scale, see S1), I would want to see a bit more work verifying that the scale and multi-lingual nature of the dataset is a useful feature:

W1a) For scale, it would be helpful to verify that the problems and tests are high quality and consider automatically generating more tests, given that work like CodeContests [Li et al.] find that many naturally-occurring tests have spotty coverage, leading to false positives.

W1b) For multi-lingual, it would help to do more analysis of difference in model performance across languages, perhaps as a function of the data the models were trained on (for open-source models with known training data, like StarCoder).

W2) The analysis and model evaluation was a bit thin:

W2a) The experiments consisted mainly of ChatGPT with some additional experiments on open-source models. While I appreciated the use of open-source models, they were relatively small: CodeLlama-Instruct (up to 13B) and StarCoder (3B, fine-tuned) models, and given the very low performance of these models, and that a different model family was fine-tuned than used zero shot, I don't know that much can be drawn from the finding that StarCoder-3b fine-tuned outperformed CodeLlama-7b instruct.

W2b) I didn't feel that the temperature analysis or "reasoning spectrum" added much to the paper, as it was difficult to find a clear takeaway from them.

W2c) The other tasks beyond program synthesis were under-explored. Code translation and APR, in particular, seem potentially useful to me (in particular, I think it's exciting that APR is based on chronological submissions by a participant, as explained in E.5), but both of these had very limited experimental results. The difficulty of the retrieval task also seems to depend heavily on the size of the retrieval dataset (as evidenced in the text description of Table 4, with high vs low resourced languages), and I think future work here might need to introduce other metrics or account for the size of the datasets in some way.

W2d) I was curious about the distribution of tags in the dataset, as this will give a lot of information about the fine-grained types of algorithms that are involved.

W3) The writing of the paper could be improved. In particular, I'm afraid I didn't fully understand the motivation for or implementation of the data balancing method in section 2.1. I think that this might be better suited to the appendix, with the main text focusing on addressing some of the weaknesses above (e.g. additional experiments, or evaluation), or moving some of the appendix text on the Task Construction process to the main text.

**Questions:**

Q1)  I was confused by the \sum_{v \in V} f(u, v) = 0 in section 2.1, since the range of the flow is only non-negative integers. Does the direction of the edges negate some edge weights in the sum?

Q2) How do the top-k retrieval accuracy metrics account for there being multiple solutions in a given language per problem (if I understand correctly)?

Other clarification points (not necessary to answer in the author response):
- In the abstract, specify the source of the data (code contests problems)
- "parallelism of multilingual data" in the intro was unclear; might convey parallel programming.
- "Evaluation and its granularity" section of the intro was unclear about what global level meant or how to produce unit tests for it.
- The description in section 2.1 had a lot of detail about the algorithm, but I felt it would be better served to give some high-level intuition instead. It was unclear to me why the method does val/test division before filtering (as filtering might affect the number of samples in val and test).
- Figure 3 is hard to read since C and Rust have the same color.
- Is Rust the only rare language? What about e.g. Go and Kotlin?
- The description of "reasoning path" on page 8 was confusing, since it only evaluates the output of the code (PASSED vs WRONG ANSWER, etc), but to me "reasoning" conveys different algorithms or code implementations that all can produce PASSED solutions.

---

> ### Author Response · Authors · 2023-11-20
> **Review response**
>
> We thank the reviewer for the review. Here is our response.
>
> **W1)** Please see our General response on Novelty. In addition to that, we provide 62,798 unit tests (Table 2), more than double the number in previous benchmarks. These tests comprehensively cover all necessary scenarios to verify the correctness of a solution. The correctness, defined by a set of unit tests, is initially confirmed by a human annotator/problem setter when the problem is created on Codeforces and later validated by contestants and the problem-solving community. Thus the problems and unit tests are of high quality. We have also  ensured that our evaluation data includes the full unit tests, a step not taken in the CodeContest and APPS benchmarks. We believe this addresses a major shortfall in previous benchmarks and is rectified in ours. XCodeEval can be a good resource to train models for automatic test case generation, which is not within the scope of this (already dense) benchmark paper.
>
> **W2a)** As of now, one of the main issues with the new genre of Large Language Models (LLMs) is that not all models are open-source. The way right now open-source and closed-source models “communicate” is via evaluation benchmarks. While open-source models are a significant part of the ecosystem, our proposed benchmark aims to ensure its applicability to any black-box (closed-source) model. The objective of the experiments presented in Table 5 is to demonstrate the impact of the finetuning data given along with the benchmark. The experiment in Table 5 reveals that a fine-tuned version of the 3B model can outperform a general-purpose 7B SFT model, but it cannot surpass a 13B general-purpose SFT model. Also this experiment highlights the trade-offs between model size and fine-tuning on specific corpora.
>
> **W2b)** Temperature is a critical hyperparameter in the decoding strategy of language models. Analysis involving temperature demonstrates a significant performance gap when decoding at different temperature values. Given that generating programs necessitates maintaining consistent reasoning in long-range sequences, we illustrate that model developers should consider this parameter attentively during evaluation.
>
> The reasoning spectrum provides a singular visualization across the entire evaluation benchmark, comparing unit tests across languages. For instance, a vertical green line on the y-axis indicates that all languages can pass that particular unit test. Additionally, the reasoning spectrum offers a more detailed comparison of unit tests between languages.
>
> **W2c)**  Please refer to Table 3 for the complete results of the APR and Code-Translation Task. Due to spacing issues, a detailed description of all the tasks is also provided in Appendix E.
>
> **W2d)** The distribution of tags can be found in Appendix E.2. Additionally, the effect of our data selection strategy, as explained in Section 2, is detailed in Table 7.
>
> **W3)** Let's assume that you have 25 million samples from 7,500 unique problems in 11 programming languages across 31 domains (where 'domains' refers to tags). This presents a unique combinatorial challenge in selecting data for training, development, and test distribution. In Section 2, we propose a novel data splitting and selection schema based on the geometric mean and graph-theoretic principles.
>
>
> **Q1)** Yes, since we use Dinic's algorithm to implement circulation flow with upper and lower bounds, the direction of the edges negates some edge weights in the sum. Thank you for pointing this out. To account for the full simulation and the backward flows, we will revise the our text  $f: E\longrightarrow\mathbb{Z}+$  to $f: E\longrightarrow\mathbb{Z}$ for better understanding of the context. However, this is just to re-iterate that $f: E\longrightarrow\mathbb{Z}+$ holds true for all the potential sets of flow to be selected as a solution. In contrast, $f: E\longrightarrow\mathbb{Z}-$ is necessary for the internal computations of Dinic's algorithm.
>
>
> **Q2)** In our context, Top-k accuracy determines whether a PASSED solution exists within the top-k retrieval of the code for a query problem ID. We adopt the standard interpretation of retrieval accuracy in NLP and IR literature which denote the fraction of queries for which the top-k retrievals contain any of the corresponding solution/answer code. We refer to [BEIR](https://openreview.net/forum?id=wCu6T5xFjeJ) and [DPR](https://arxiv.org/abs/2004.04906) for more details.
>
>
> [1] Muennighoff, Niklas, et al. "Scaling Data-Constrained Language Models." arXiv preprint arXiv:2305.16264 (2023)

---

> > ### Author Response · Authors · 2023-11-20
> > **Cont. Review response**
> >
> > **Other Clarification**:
> > - **Source of the data**: The data was sourced from codeforces.com. We politely crawled the data over a period of six months. We did not use any data from code-contest or other sources.
> > - **`Global` Meaning**: By 'global,' we mean that the code is a complete program that takes standard input and outputs to standard output. Recently, most of the benchmarks (but not all) have been proposed for module-level evaluation. By 'module,' we mean that a function is written for evaluation.
> >  - The description of "reasoning path" on page 8 was confusing. The program synthesis problem is evaluated through a set of unit tests. This collection of unit tests ensures coverage of all conditions required by the algorithms, including corner cases. Assume an algorithm necessitates five concepts of reasoning. A problem setter might implement these five concepts through 50 distinct unit tests. Therefore, a single unit test or a collection of them can define one or several reasoning paths.

---

### Official Review · Reviewer_LKoY · 2023-11-07

**Soundness:** 3 good
**Presentation:** 2 fair
**Contribution:** 2 fair
**Rating:** 3
**Confidence:** 4

**Summary:**

The paper introduces xCodeEval, a large-scale, multilingual, multitask benchmark designed to evaluate code language models on ode understanding, generation, translation and retrieval.  The benchmark includes 25M coding examples from about 7.5K unique problems in 11 programming languages. It employs execution-based evaluation with both training and test sets. The authors conduct various benchmarking on leading models and show that xCodeEval presents a significant challenge to the current state of LLMs.

**Strengths:**

- The benchmark covers a wide range of code tasks.
- The docker-based execution-based evaluation can streamline the evaluation across platforms and potentially compliment many existing widely-used lexical-match-based benchmarks.
- The authors presented various anaylsis and clearly discussed limitations and risks, which are crucial for benchmarks.

**Weaknesses:**

- While the paper appears to be comprehensive, it is extremely dense and not self-contained. It appears that the authors aimed at covering a ton of work and have to skip most details, resulting in most sections being cursory. Figures and tables are not presented clearly and barely readable. What's worse, the related work section is completely missing in the main body of the paper, which is unacceptable. The authors should consider how to streamline the content and make sure that the main paper is self-contained.

- The novelty is limited. The authors use codeforces as the source dataset. On one side, it is similar to various existing code-competition execution-based benchmarks from HumanEval and MBPP to APPS, CodeContests, DS-1000, MBXP, HumanEval-X, MultiPL-E, BabelCode, etc. which cover a wide range of problems in different level of difficulty and/or multiple languages. Among all, xCodeEval overlaps a lot with CodeContests which was partially built on Codeforces too. On the other side, it is unclear how the rest of tasks, e.g., code retrieval, can benefit the current set of evaluations. The authors should clearly articulate the novelty of the work.

- The dataset comes with finetuning data, however, it wasn't explained well why we need finetuning data at the era of LLMs. How would it add value in benchmarking an LLM?

- The license of the dataset is CC-BY-NC, which significantly limits the usability of the dataset, especially given most LLMs come from the industry. Is this coming from the authors or codeforces? If the former, it would be great if the authors can re-consider the decision.

**Questions:**

See Weaknesses for questions. In addition,

- How does xCodeEval ensure the quality of its benchmarks across different languages?
- How would the new split algorithm (sec 2.1) improve over the baselines (e.g., random selection and/or time-based selection)?
- Did the authors obtain hidden tests (if any) from codeforces?

**Details Of Ethics Concerns:**

It is unclear whether the authors have the right to create the dataset from crawling codeforces.

---

> ### Author Response · Authors · 2023-11-20
> **Review response**
>
> We thank the reviewer for his comment. Here we respond to your comments and concerns.
>
> **On the Clarity of the paper (Dense contents)**: In order to demonstrate the remarkable aspects and distinctive features of xCodeEval and our contribution, it is essential to depict the bottlenecks of the existing code benchmarks, how xCodeEval addresses them, our novel data selection schema based on a “Circulation problem with lower and upper bound strategy”, developing a full NL-Code benchmarking peipeline including our multilingual code execution engine, ExecEval, presenating data properties and uniqueness, all featured tasks, a comprehensive analysis and ablation study on the reasoning capabilities of LLM models considering the possible data leakage which is really a lot of contents to present.  Unfortunately, ICLR has a page limit. In Revision, we will improve the clarity of the paper and leverage the addition page. However we believe that in the current context, our paper is self-contained in terms of benchmark attributes, evaluation and analysis.
>
> **On the discussion of Novelty**:  Please follow the general response for the Novelty Description. In our General reply, we explain how xCodeEval differs from other datasets and its unique properties.
>
> **Fine-Tuning Dataset**: We disagree with the claim that with LLs, the finetuning is no longer needed, rather finetuning models still prevail especially for adaptation to specific code-tasks or domains, enhancing performance, protecting data privacy and allowing benchmarking on code-specific tasks. It enables customization, addresses model analysis and so on. Nonetheless, the fine tuning dataset is specially intended to be used for finetuning smaller models considering that the evaluation benchmark may not be suited for zero-shot evaluation of such models (at least in today’s standard). In Figure 5, we show that fine-tuning a 3B model with training data can outperform a 7B general purpose SFT model. This shows that the training data can be used for enabling document level executability to smaller models. We also take extra precaution for train-dev-test overlap (Section 2.1) so that the integrity of the benchmark remains robust and training/finetuing data doesn’t interfere with benchmark evaluation. Also open sourcing fine tuning data helps the model evaluators to do data contamination analysis.

---

> > ### Author Response · Authors · 2023-11-20
> > **Cont: Review response**
> >
> > **License Issue**:
> >
> > We want to emphasize that codeforces data already exists in popular benchmarks; CodeContest [1], APPS [2] and Avatar [3] use codeforces data for training and evaluation. We strongly believe that the license should not be a weakness factor for the paper as it does not restrict any research usage (as per the ICLR policy).
> >
> >
> > **How does xCodeEval ensure the quality of its benchmarks across different languages?**: We take the following measures to make sure the quality of the benchmark.
> >   - One of the main attributes of the generative evaluation of xCodeEval is that it is language agnostic. The output is evaluated by the unit tests. We make sure that our evaluation framework ExecEval can properly evaluate the languages that we advertise. We also add safety features using “prlimit” and “seccomp” at the time of code execution.
> >   - All the problems in the codeforces do not have a complete set of test cases. Thus the previous benchmarks CodeContest [1] and APPS [2]  have incomplete unit test benchmarks. We make sure that all of our validation/test problems have the complete set of unit tests. More details are in the General Reply and “Evaluation and its granularity” section (Page 3) of the paper.
> >   - We make sure that validation-test distribution has a balanced distribution over the domain of the problems. We proposed a novel approach to create a balanced validation-test distribution using circulation problems with upper and lower bound in section 2.
> >   - We have cleaned the corpus with automated tools  identifying and removing codes with sensitive details (i.e., email address), resulting in the removal of approximately 2 million samples from our original collection. We also followed all the necessary steps mentioned by Gebru et al [8], Holland et al [9], Hutchinson et al [10].
> >
> >
> > **Performance of the new split algorithm**: Section D2 with Table 7 in the Appendix shows that the new split algorithm (Sec 2.1) improves from the random baseline. We measured the skew, and standard deviation of the distribution of tags and showed that our proposed algorithm performs better than random data selection.
> >
> > **Unit test details**: Yes, all the hidden test sets are obtained from codeforces which are openly available. For validation and test datasets, we ignore the problems for which the test cases are incomplete. A complete collection of test sets ensures the correctness of a proposed solution..
> >
> > We believe that we have addressed all the issues raised by the reviewer. We are open to discuss and engage with the reviewer for further discussion. We urge the reviewer to raise any concerns on our replies. We strongly believe that with a proper, engaging discussion we can come to a better conclusion and clear all the confusion and miscommunication. We wait for the reviewer's response.
> >
> > [1] Li, Yujia, et al. "Competition-level code generation with alphacode." Science 378.6624 (2022): 1092-1097.
> > [2] Hendrycks, Dan, et al. "Measuring coding challenge competence with apps." arXiv preprint arXiv:2105.09938 (2021).
> > [3] Ahmad, Wasi Uddin, et al. "Avatar: A parallel corpus for java-python program translation." arXiv preprint arXiv:2108.11590 (2021).
> > [4] https://pilehvar.github.io/wic/
> > [5] https://github.com/facebookresearch/anli#license
> > [6] https://github.com/facebookresearch/XNLI
> > [7] Gebru, Timnit, et al. "Datasheets for datasets." Communications of the ACM 64.12 (2021): 86-92.
> > [8] Holland, Sarah, et al. "The dataset nutrition label." Data Protection and Privacy 12.12 (2020): 1.
> > [9] Hutchinson, Ben, et al. "Towards accountability for machine learning datasets: Practices from software engineering and infrastructure." Proceedings of the 2021 ACM Conference on Fairness, Accountability, and Transparency. 2021.

---

### Official Review · Reviewer_P8wu · 2023-11-08

**Soundness:** 3 good
**Presentation:** 2 fair
**Contribution:** 2 fair
**Rating:** 5
**Confidence:** 5

**Summary:**

The paper presents xCodeEval, a benchmark designed for evaluating code generation models. This benchmark, notable for its size and scope, encompasses over 25 million coding examples from approximately 7,500 unique problems and extends support to 11 programming languages. A key feature of xCodeEval is its emphasis on execution-based evaluation, operationalized through the introduction of ExecEval—a multilingual code execution engine that uses unit tests to evaluate code in all supported languages. The authors test the benchmark with several pre-trained language models.

**Strengths:**

1. The paper introduces a large-scale, execution-based benchmark, xCodeEval, filling a gap in existing evaluation protocols that often rely on smaller datasets (HumanEval, MBPP) or non-execution metrics like BLEU and exact match.

2. A standout feature of this work is the parallel, distributed execution framework, ExecEval. This innovation allows for efficient and scalable evaluation of code across multiple programming languages, which is crucial for large-scale benchmarks.

3. The detailed analysis of OpenAI's LLM performance on xCodeEval offers valuable insights. It provides a clear picture of where current LLMs excel and where they struggle.

**Weaknesses:**

1. The benchmark draws exclusively from Codeforces, which may limit the novelty of the dataset, as similar approaches have been used in other benchmarks like APPS.

2. The reliance on a single platform like Codeforces means the benchmark might not capture the full spectrum of coding tasks. Basically the benchmark is limited to algorithm contest questions.

3. The high difficulty level of the benchmark, as evidenced by low pass rates (less than 4%) across several 3B/7B/15B LLMs in Table 5, suggests it may be too challenging for all models with less than 30B parameters. This limits its utility for evaluating a broader range of model sizes and capabilities.

4. There are concerns about data contamination when evaluating models such as GPT-3.5, which have been trained on extensive web data that may include the very solutions from Codeforces used in the benchmark, potentially skewing the results in favor of GPT-3.5. This is because xCodeEval contains questions from 2010 - 2022. It would be interesting to see whether GPT-3.5 has consistent performance on questions after 2022.

5. The visual presentation of the paper could be enhanced for better clarity and accessibility. Specifically, some graphics and tables, such as the legend in Figure 4, are difficult to read in print form.

In summary, while the paper contributes a valuable tool for advancing code LLMs, these aspects should be addressed to fully realize its potential.

**Questions:**

It would be interesting to see whether GPT-3.5 has consistent performance on questions after 2022.

---

> ### Author Response · Authors · 2023-11-21
> **Review response**
>
> We thank the reviewer for his comment. Here we respond to your comments and concerns.
>
> ## The benchmark draws exclusively from Codeforces ...
> There are many fundamental issues we solve that exist in prior evaluation frameworks like CodeContest and APPS. Here are some description (also include in the general response and paper)
>   - **How xCodeEval differs from APPS, CodeContests**: For CodeContests ([repository](https://huggingface.co/datasets/deepmind/code_contests/viewer/default/test?row=1)), 44 out of 165 test samples had no private unit tests. Additionally, there were inconclusive test cases. For instance, consider [this](https://codeforces.com/contest/1575/submission/130567445) problem in row 1 which contains 7 private test cases, while in the original codeforces competition, there are 34 test cases (accessible through the link here, you might need to log in and then click on Click to see test details), and some of them were incomplete (ended by ...) even in the codeforces API. We excluded problems with incomplete test cases from our development and test datasets. It's important to note that a problem is considered solved only when all the unit test cases pass successfully. We are grateful to our insightful reviewer for bringing up this aspect, and we plan to provide more context on these statistics in the final version of our paper. We mention that in “Evaluation and its granularity” section (Page 3).
> Similar problem occurs with APPS. For example: https://huggingface.co/datasets/codeparrot/apps (test split, row index 6) and codeforces source https://codeforces.com/contest/1217/problem/B, There are 365 test cases, while APPS included only 224. Note that these test cases, created by expert humans, as a whole ensure the correctness of a solution: a solution passed on all except one is considered as a wrong answer. xCodeEval ensures that this hypothesis preserves, while both APPS and CodeContest don’t ensure that.
>  - APPS and CodeContests offer only program synthesis tasks. xCodeEval proposes  5 different tasks from Classification, Generation and Retrieval tasks. From Table 1, APPS has  22,711 and  27,220 unit tests while xCodeEval covers 62,798 unit tests.
>  - APPS and CodeContests problems do not provide some metadata, while xCodeEval provides the following crucial metadata:
>    - **Timestamp**: Problem release date which helps to analyze data contamination (check Fig 4-left). This is indeed very crucial see this interesting [blog](https://lmsys.org/blog/2023-11-14-llm-decontaminator/)  and [paper](https://arxiv.org/pdf/2311.04850.pdf) on beating GPT-4!
>    - **Difficulty level**: Human annotated difficulty level in the range 800 to 3500).  APPS problems are divided into Competition, Interview and Introductory These are more like domains. For example, an Introductory problem can be harder than a Competition level problem. For example, in [APPS competition test, row 5](https://huggingface.co/datasets/codeparrot/apps/viewer/competition/test?p=9&row=904) has a difficulty of 1300 whereas in [APPS introductory test row 3](https://huggingface.co/datasets/codeparrot/apps/viewer/introductory/test) has a difficulty of 2100 at codeforces. Both of these diffuculty levels are annotated by human experts.
>    - **Tags/category**: xCodeEval comes from 31 different domains. Each of the domain problems can be identified during evaluation to understand about the domain biases of the code generation in LLM; see Figure 8 for details.
>
> ## The reliance on a single platform like Codeforces means the benchmark might not capture ...
> In our General Response, we explain xCodeEval’s novelty and how it differs from other datasets. In addition, Table 6 shows that most of the datasets typically cover a single domain. Specifically, the popular module-level benchmarks like HumanEval, MBPP, MBXP, HumanEval-X, MultiPL-E and BabelCode evaluate language specific features which usually exist in book exercises. The DS-100, Exe-DS, ARCADE are mainly python-dependent execution based Data Science problems. Other than that, most of the competitive programming language based evaluation benchmarks [1][2][3][4][5] use problems from ICPC style problems. Though xCodeEval primarily uses codeforces problems, it has the largest ICPC style problem collection. In addition to the regular contest, codeforces regularly host contests for graduate university, past ICPC contests, educational rounds (traditional problems) as well as [legacy problems](https://codeforces.com/gyms). We want to emphasize that though codeforces sounds like a single site, it hosts the largest variety of problems which covers more than 31 domains of problems.

---

> > ### Author Response · Authors · 2023-11-21
> > **Cont. Review response**
> >
> > ## The high difficulty level of the benchmark, as evidenced by low pass rates (less than 4%) across several 3B/7B/15B LLMs ...
> > Please see our general response. We strongly believe that the high difficulty level is indeed a strength instead of weakness. With the release of ChatGPT and GPT-4, it is evident that most of the benchmarks are either already memorized by the LLMs from the training data or the LLMs are generalized enough so that the benchmark becomes too easy. See this interesting blog  and paper why we need to rethink benchmarking and contamination for LLMs.
> > We believe that a verifiable robust benchmark like xCodeEval can enable evaluation on the current genre of LLM as well as the next genre of more capable frontier models. (i) reasoning (ii) long-range document level multi-lingual seven code tasks (iii) following extreme sophisticated details and (iv) accounting for smallest errors that result in inaccuracy, and (v) a comprehensive analysis of model’s performances considering possible data leakage.
> >
> > ## The visual presentation of the paper could be enhanced ...
> >
> > We will improve the visual presentation of the paper.
> >
> > ##  It would be interesting to see whether GPT-3.5 has consistent performance on questions after 2022 ...
> >
> > There are 17 problems in the test split of program synthesis dataset which are from or after 2022 with difficulty ranging from 800 to 3200 where pass@5 score for c++ was 1.47% and that of python was 0% with an average of 2% and 2.13% for versions 0613 and 0301 of gpt3.5-turbo . Following Figure 4a, this does indeed indicate a high level of memorization in GPT-3.5.
> >
> >
> > | Model/API |  C | C# | C++ | Go | Java | Javascript | Kotlin | PHP | Python | Ruby | Rust | Avg |
> > |---|---|---|---|---|---|---|---|---|---|---|---|---|
> > | gpt-3.5-0301 | 1.47 | 3.53 | 4.23 | 1.47 | 5.39 | 5.88 | 0 | 1.47 | 0 | 0 | 0 | 2.13 |
> > | gpt-3.5-0613 | 1.47 | 0  | 1.47 | 5.26 | 1.47 | 1.47 | 2.8 | 5.12 | 0 | 1.47 | 1.47 | 2.0 |

---

### Official Review · Reviewer_Hh2i · 2023-11-11

**Soundness:** 3 good
**Presentation:** 2 fair
**Contribution:** 2 fair
**Rating:** 5
**Confidence:** 4

**Summary:**

The paper presents work done to enhance code LLM evaluation abilities, through a new benchmark that supports additional languages, includes multiple tasks for which code LLMs are being used today and additional framework to enable automatic evaluation through code execution.

**Strengths:**

Code LLMs are now mainstream and to go beyond small interactive code snippets, we need reliable evaluation mechanisms, especially those using code's advantage over other text - verifiable executability. This work furthers the body of paired samples enhanced with useful metadata which helps support multiple tasks. The work further validates the thesis by trying relatively smaller SoTA models like *Coder-base and Llama all the way to OpenAI's offerings.

**Weaknesses:**

While the work is important and highly relevant, the contribution feels incremental:
- there are plenty of enhancements coming into code datasets with additional processing of github based datasets like Stack v2
- the additional languages are also mostly mainstream and available in different datasets; it would have helped if we add really low resource languages (Stack v2 has shown that too with even languages like COBOL)
- execution based evaluation has been around for multiple years now, especially unit tests, The distributed execution as well as additional metadata is a great value add, but it's been meshed with other features in one paper - it might have been better to split these into separate submissions to be able to evaluate each on its own merit
- execution oriented evaluation also has its limitations and can lead to incorrect code being validated correctly especially if there's no return value or booleans; it doesn't help sufficiently with secure code generation or deprecated API etc.

Maybe the issue is with combining several concepts into one paper which limits the amount of presentation you can provide to each of the concepts - separating benchmark, execution framework, and multiple languages might have helped at least one to rise sufficiently.

**Questions:**

- What would it take to expand the work to include other low resource languages including from a different domain?
- Since you used StarCoderBase as one of comparison points, could you provide value differentiation of the new benchmark over the github content in Stack used to train it? One would expect most internet code to also be available in github and so all of the models used to compare might have been exposed to most of this code.
-

**Details Of Ethics Concerns:**

The biases inherent in datasets would show in the various tasks as well. Analyzing this relatively small dataset for that would be helpful.

---

> ### Author Response · Authors · 2023-11-20
> **Comparison with stack & programming language availability**
>
> We thank the reviewer for his comment. Here we respond to your comments and concerns.
>
> # Comparison with Stack V2 and the Objective of the paper:
>
> We want to emphasize that the objective of **Stack v2** and **xCodeEval** dataset is completely different thus they cannot be compared with each other. Stack v2 is a pre-training (usually via self-supervised) corpus while xCodeEval is an (NL-Code, Code-Code annotated) evaluation benchmark taken from 25M samples using a novel data selection strategy. It comes with the largest set of languages (see Table 1 & Table 6 ) compared with other benchmarks. It also provides an additional training corpus which can be used to do fine tuning. The training corpus is specially required to enable executability to the smaller models.
>
> # What would it take to expand the work to include other low resource languages including from a different domain?
> Since xCodeEval is a unit test based benchmark, the program synthesis and code translation tasks of xCodeEval can be extended to any number languages as long as the evaluation framework supports that language. The APR task can be modified to Cross-Lingual Program-repair task and extended to any number of languages. For retrieval and classification tasks, we need data for low resource languages to extend the evaluation framework. However, these tasks can also be used for contrastive representation learning.
>
> # Since you used StarCoderBase as one of comparison points, could you provide value differentiation of the new benchmark over the github content in Stack used to train it?
> Again, we want to highlight that Stack is a large corpus intended to perform large-scale pretraining, while xCodeEval is an evaluation benchmark. We also include a fine-tuning corpus to facilitate fine-tuning of smaller models that don’t have good document level executability. We use StarCoderBase to showcase this hypothesis. Please also follow the General response to get a good idea how xCodeEval differentiates from other benchmarks.

---

### Author Response · Authors · 2023-11-20
**General Response**

# On the question of Novelty

1. **Novel Data Selection Strategy**: xCodeEval proposes a novel data selection strategy using a “Circulation problem with lower and upper bound strategy”.
2. **How xCodeEval differs from HumanEval, MBPP, MBXP, HumanEval-X, MultiPL-E, BabelCode**: We want to emphasize Table 1 and Table 6 to understand the novelty and distinction of our benchmark. HumanEval, MBPP, MBXP, HumanEval-X, MultiPL-E, BabelCode are module (or function) level benchmarks, while the tasks in xCodeEval mainly offer document level evaluation. In addition, we want to emphasize the evaluation size of the benchmark. Current genres of language models are often trained with more than a 1T tokens. HumanEval, HumanEval-X, MultiPL-E, BabelCode all contain less than 164 samples. MBXP, MBXP contains 900 samples but provides 1,500 unit tests on language specific tasks. In contrast, xCodeEval is a language agnostic benchmark, consisting of 952 - 74,733 evaluation samples (based on different tasks) with 62,798 unit tests.
3. **How xCodeEval differs from DS-1000**: DS-1000 is exclusively based on a Data Science problem consisting of 1000 tests only on Python. xCodeEval supports 11 languages and can be evaluated on any language as long as the compilation is integrated in ExecEval.
4. **How xCodeEval differs from APPS, CodeContests**: While both APPS and CodeContests feature codeforces problems, they have major issues on completeness of the Unit Tests.
  - For CodeContests ([repository](https://huggingface.co/datasets/deepmind/code_contests/viewer/default/test?row=1)), 44 out of 165 test samples had no private unit tests. Additionally, there were inconclusive test cases. For instance, consider [this](https://codeforces.com/contest/1575/submission/130567445) problem in row 1 which contains 7 private test cases, while in the original codeforces competition, there are 34 test cases (accessible through the link here, you might need to log in and then click on Click to see test details), and some of them were incomplete (ended by ...) even in the codeforces API. We excluded problems with incomplete test cases from our development and test datasets. It's important to note that a problem is considered solved only when all the unit test cases pass successfully. We are grateful to our insightful reviewer for bringing up this aspect, and we plan to provide more context on these statistics in the final version of our paper. We mention that in “Evaluation and its granularity” section (Page 3).
Similar problem occurs with APPS. For example: https://huggingface.co/datasets/codeparrot/apps (test split, row index 6) and codeforces source https://codeforces.com/contest/1217/problem/B, There are 365 test cases, while APPS included only 224. Note that these test cases, created by expert humans, as a whole ensure the correctness of a solution: a solution passed on all except one is considered as a wrong answer. xCodeEval ensures that this hypothesis preserves, while both APPS and CodeContest don’t ensure that.

- APPS and CodeContests offer only program synthesis tasks. xCodeEval proposes  5 different tasks from Classification, Generation and Retrieval tasks. From Table 1, APPS has  22,711 and  27,220 unit tests while xCodeEval covers 62,798 unit tests.

- APPS and CodeContests problems do not provide some metadata, while xCodeEval provides the following crucial metadata:
  - **Timestamp**: Problem release date which helps to analyze data contamination (check Fig 4-left). This is indeed very crucial see this interesting blog  and paper on beating GPT-4!
  - **Difficulty level**: Human annotated difficulty level in the range 800 to 3500).  APPS problems are divided into Competition, Interview and Introductory These are more like domains. For example, an Introductory problem can be harder than a Competition level problem. For example, in APPS competition test, row 5 has a difficulty of 1300 whereas in APPS introductory test row 3 has a difficulty of 2100 at codeforces.
  - **Tags/category**: xCodeEval comes from 31 different domains. Each of the domain problems can be identified during evaluation to understand about the domain biases of the code generation in LLM; see Figure 8 for details.

To summarize, xCodeEval is the first (i) Multilingual, (ii) Document level, (iii) Complete Unit test covering correctness of the problem, (iv) Diverse in terms of task type (Classification, Generation, and Retrieval), and (v) Metadata enabled benchmark. The benchmark not only provides evaluation data but also provides a general purpose dockerized evaluation framework for compilers. While the prior benchmarks are impactful but often not complete, we believe the xCodeEval will fill the important gaps between many different benchmarks and stay robust not only for current genre of models but also for the Frontier Models.

---

> ### Author Response · Authors · 2023-11-20
> **Why this benchmark Matters**
>
> In Section 1, we summarize the bottlenecks of existing datasets and benchmarking systems to evaluate the code generation capabilities of recent generative LLMs and how xCodeEval addresses those limitations specifically by filing the evaluation gaps of (i) reasoning (ii) long-range document level multi-lingual seven code tasks (iii) following extreme sophisticated details and (iv) accounting for smallest errors that result in inaccuracy, and (v) a comprehensive analysis of model’s performances considering possible data leakage. With the release of ChatGPT and GPT-4, it is evident that most of the benchmarks are either already memorized by the LLMs from the training data or the LLMs are generalized enough so that the benchmark becomes too easy. See this interesting [blog](https://lmsys.org/blog/2023-11-14-llm-decontaminator/)  and [paper](https://arxiv.org/pdf/2311.04850.pdf) why we need to rethink benchmarking and contamination for LLMs. We believe that a verifiable robust benchmark like xCodeEval can enable evaluation on the current genre of LLM as well as the next genre of more capable frontier models.

---

### Meta-Review · Area_Chair_9s2S · 2023-12-11

**Metareview:**

This paper introduces a multi-programming-language code dataset based on codeforces.com. The dataset includes 17 languages, with 7.5K problems, where problems have solutions in multiple languages. The problems also have test cases, 63K in total, which can be used in code generation tasks to verify the correctness of model-generated solutions. This is more than previous datasets and this work introduced a Docker-based execution testing environment, ExecEval.

Reviewers generally felt high quality evaluations is a contribution to the community. They liked that this dataset included a larger set of programming languages, the detailed analysis especially regarding time-cut off, and appreciated the convenience from ExecEval. However all reviewers questions the novelty and added value of this particular dataset. Several reviewers pointed out execution based evaluation is already mainstream in the area and the collection method is similar to APPS.  Reviewers also complained that the authors tried to do too much in the paper which resulted in confusion and not enough experiments to sufficiently establishing the added value of the proposed benchmark over existing but smaller evaluations. Reviewers were also confused about selection methods and question some decisions that may impact the quality of the dataset. This is the most important negative point. If bigger benchmark set give you the same conclusion as a smaller benchmark, then size is not a benefit in itself. One standard is for the authors to show where one would draw the wrong conclusions from previous benchmarks and that the conclusion from the proposed benchmark is indeed correct.

The authors corrected a few misconceptions during the rebuttal, such comparing to pretraining dataset, and seeing low accuracy as an disadvantages. These are all correct technical points by the author but are not the main issues raised by reviewers.

For non-technical points, the authors included their github URL in the draft itself and one reviewer objected that CC-BY-NC is too restrictive.

**Justification For Why Not Higher Score:**

reviewer majority

**Justification For Why Not Lower Score:**

N/A

---

### Decision · Program_Chairs · 2024-01-16

Reject